# Giant gate-controlled odd-parity magnetoresistance in one-dimensional channels with a magnetic proximity effect

Kosuke Takiguchi [1], Le Duc Anh [1,2,3,4] ✉, Takahiro Chiba[5], Harunori Shiratani[1], Ryota Fukuzawa[1,6], Takuji Takahashi[6,7] & Masaaki Tanaka [1,4,7] ✉

According to Onsager's principle, electrical resistance $R$ of general conductors behaves as an even function of external magnetic field **B**. Only in special circumstances, which involve time reversal symmetry (TRS) broken by ferromagnetism, the odd component of $R$ against **B** is observed. This unusual phenomenon, called odd-parity magnetoresistance (OMR), was hitherto subtle (< 2%) and hard to control by external means. Here, we report a giant OMR as large as 27% in edge transport channels of an InAs quantum well, which is magnetized by a proximity effect from an underlying ferromagnetic semiconductor (Ga,Fe)Sb layer. Combining experimental results and theoretical analysis using the linearized Boltzmann's equation, we found that simultaneous breaking of both the TRS by the magnetic proximity effect (MPE) and spatial inversion symmetry (SIS) in the one-dimensional (1D) InAs edge channels is the origin of this giant OMR. We also demonstrated the ability to turn on and off the OMR using electrical gating of either TRS or SIS in the edge channels. These findings provide a deep insight into the 1D semiconducting system with a strong magnetic coupling.

Investigation of new magnetoresistance (MR) phenomena is an important issue in condensed matter physics, magnetism, and spintronics. For example, the discovery of giant MR[1,2] and tunneling MR[3,4] paved the way to the creation of non-volatile storage and memory devices. Generally, these MRs are even functions of external magnetic field $B$ according to Onsager's principle[5]. However, it may not be the case when time reversal symmetry (TRS) is broken by magnetism in the system. The odd-parity MR (OMR) in a linear-response regime has been observed in systems where TRS is violated[6–9]. (See also Supplementary Table 1). To explain these OMR phenomena, various possible origins were proposed, including non-trivial Berry curvature, magnetic moments, side jump mechanism[10], and coexistence of spin–orbit

interaction (SOI) and ferromagnetic coupling in a helical magnet[11]. Even in such rare systems, the OMR magnitude is typically very subtle (the magnitude reported thus far is at most 2%). In addition, these systems reported thus far are metallic, which hinders the control of OMR by external means such as electrical gate voltage.

In this Article, we report a giant and gate-controlled OMR in the edge transport channels of an InAs thin film interfaced with a ferromagnetic semiconductor (FMS) (Ga,Fe)Sb[12–14] layer (see Fig. 1a). The OMR is found to be unprecedently large; the resistance change is 27% of the total resistance when the **B** direction is reversed between ±10 T at $I = 1\,\mu A$. This is striking, considering that the SOI of InAs is much smaller than other materials such as $SmCo_5$ and pyrochlores in which OMR was

[1]Department of Electrical Engineering and Information Systems, The University of Tokyo, Bunkyo-ku, Tokyo 113-8656, Japan. [2]Institute of Engineering Innovation, The University of Tokyo, Bunkyo-ku, Tokyo 113-8656, Japan. [3]PRESTO, Japan Science and Technology Agency, Kawaguchi, Saitama 332-0012, Japan. [4]Centre for Spintronics Research Network, The University of Tokyo, Bunkyo-ku, Tokyo 113-8656, Japan. [5]National Institute of Technology, Fukushima College, Iwaki, Fukushima 970-8034, Japan. [6]Institute of Industrial Science, The University of Tokyo, Meguro-ku, Tokyo 153-8505, Japan. [7]Institute for Nano Quantum Information Electronics, The University of Tokyo, Meguro-ku, Tokyo 153-8505, Japan. ✉e-mail: anh@cryst.t.u-tokyo.ac.jp; masaaki@ee.t.u-tokyo.ac.jp

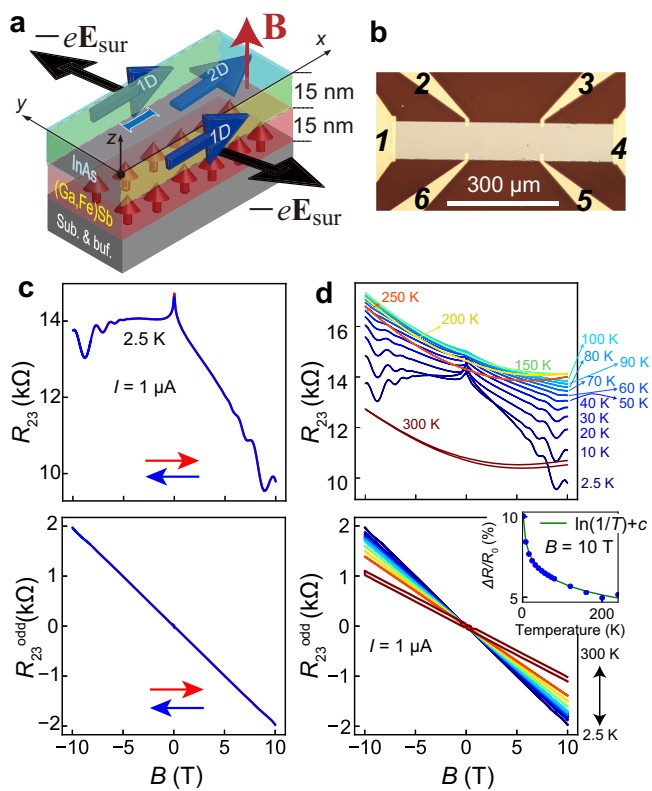

**Fig. 1 | Magnetoresistances (MRs) of InAs/(Ga,Fe)Sb bilayer heterostructures.**
**a** Schematic illustration of the InAs/(Ga,Fe)Sb heterostructure with 1D transport channels at the side edges. We applied an electric current **I** parallel to the $x$-axis and an external magnetic field **B** parallel to the $z$-axis. Because (Ga,Fe)Sb is insulating, electron carriers flow only in the InAs QW layer, both in the 2D channel and the 1D channels at the edges. The triangular potentials at the side surfaces create static electric fields **E**$_\text{sur}$ parallel to the $y$-axis at the side edges of the InAs QW. **b** Optical microscopy top view image of the device. The terminals are labeled "1"–"6", as shown in the image. **c** (Upper panel) MR of the InAs/(Ga,Fe)Sb heterostructure of sample A, measured with a DC current of 1 μA and an external magnetic field **B** applied parallel to $z$ at 2.5 K. The blue and red arrows indicate the sweep direction of **B**. (Bottom panel) Extracted odd components of the upper panel data ($R_{23}^{\text{odd}} = [R_{23}(B) - R_{23}(-B)]/2$). **d** Temperature dependences of $R_{23}$ and $R_{23}^{\text{odd}}$ of sample A at 2.5–300 K with $I = 1$ μA. Although the even-function MR and the Shubnikov–de Haas oscillation disappear at high temperatures, the OMR component remains up to 300 K. The inset of the lower panel shows the temperature dependence of $\Delta R/R_0$, where $\Delta R = R_{23}^{\text{odd}}(10\ \text{T})$ and $R_0 = R_{23}(0\ \text{T})$ (blue circles). The green curve is the fitting result obtained using the logarithmic function $\ln(1/T) + c$ ($T$ temperature, $c$ temperature-independent parameter).

observed. We argue that this originates from the simultaneous breaking of both TRS and spatial inversion symmetry (SIS), which is entwined with both a strong magnetic proximity effect (MPE) from the underlying (Ga,Fe)Sb[15] and a Rashba SOI effect at the InAs edges. Using field-effect transistor structures, we demonstrate electrical control of the OMR by individually tuning the TRS or SIS in the system. The unprecedented strong OMR with gate controllability in mainstream semiconductors such as InAs is ideal not only for elucidating the crucial roles of the TRS and SIS breakings in solid-state physics but also for providing pathways to electronic device applications.

## Results
### Magnetoresistance and its current dependence in InAs/(Ga,Fe)Sb
The structure examined in this study consists of, from top to bottom, InAs (thickness 15 nm)/(Ga$_{1-x}$,Fe$_x$)Sb (Fe content $x = 20\%$, 15 nm)/AlSb

(300 nm)/AlAs (15 nm)/GaAs (100 nm) on semi-insulating GaAs (001) substrates grown by molecular beam epitaxy (See Fig. 1a). We utilize two samples A and B with the same heterostructure in this study (see Methods in detail). In this structure, the InAs layer is a nonmagnetic quantum well (QW) that is responsible for over 99% of the electron transport because all the other layers underneath are highly resistive[15]. (Ga,Fe)Sb is an FMS with a high Curie temperature over 300 K[12–14]. The preparation and characterization of the samples are explained in ref. 15. Due to the high crystal quality and staggered band profile at the InAs/(Ga,Fe)Sb interface, in which the conduction band bottom of InAs is at lower energy than the valence band top of (Ga,Fe)Sb, the electron wavefunction in the InAs QW significantly penetrates into the ferromagnetic (Ga,Fe)Sb layer. This induces a large MPE and spin-dependent scattering in the nonmagnetic InAs electron channel[15].

We pattern the InAs/(Ga,Fe)Sb bilayers into $100 \times 600$ μm$^2$ Hall bars with electrodes labeled "1" to "6", as shown in Fig. 1b. We drive a DC current **I** from "1" to "4" and measure the voltage differences $V_{ij} = |V_i - V_j|$ ($i, j = 1, 2, 3, 4, 5, 6$), from which we obtain the resistances $R_{ij} = V_{ij}/I$. A magnetic field **B** is applied perpendicular to the film plane (**B**//**z**). As shown in Fig. 1c, d, the **B** dependence of the four-terminal resistance $R_{23}$ measured at $I = 1$ μA shows (i) a very large odd-function MR, (ii) a large negative MR, and (iii) Shubnikov–de Haas (SdH) oscillations. The last two phenomena ((ii) and (iii)), which are even functions of **B**, are characteristics of the two-dimensional (2D) electron transport with an MPE in the InAs thin film, as thoroughly discussed in our previous work[15]. Also for (iii), the angular dependence of **B** also reveals that the SdH oscillations originate from the 2D transport (See Supplementary Fig. 1). In contrast, the large odd-function component, extracted as $R_{23}^{\text{odd}}(B)$ ($=[R_{23}(B) - R_{23}(-B)]/2$), is striking. $R_{23}^{\text{odd}}$ shows a linear dependence on $B$ with SdH oscillations (see Supplementary Note 1) over the full range of magnetic field ($|B| < 10$ T) and persists up to 300 K (lower panel of Fig. 1d). $R_{23}^{\text{odd}}(B)$ is 2.0 kΩ at $B = 10$ T and 2.5 K, corresponding to 13.5% of the total resistance, and the resistance $R_{23}(B)$ is changed by 27% of the zero-field resistance $R_{23}(0)$ upon reversing $B$ from 10 T to −10 T. This is the largest OMR observed thus far. The OMR magnitude remains almost constant in the whole range of 240 nA $< I < 100$ μA, drops suddenly to one-third of its magnitude at $I_C = \sim 200$ nA, then remains at this low magnitude when $I$ is decreased further to the lower measurable limit at 50 nA (See Fig. 2a and Supplementary Fig. 3). Even when we reverse the current direction, the OMR remains unchanged (Supplementary Fig. 4). These features indicate that the OMR presented here occurs in a linear transport regime. The reason for the sudden drop at $I_C$ is discussed in Supplementary Note 2. Also, we find that the OMR magnitude depends on the crystallographic orientation, which may be due to the non-uniformity of Fe atoms in (Ga,Fe)Sb (See Supplementary Note 3).

### One-dimensional (1D) transport in InAs and the origin of OMR
An important observation, obtained by comparing $R_{23}$ and $R_{65}$ in Fig. 2b, is that the sign of the OMR flips when we switch the voltage terminals contacting the side edge while maintaining the same measurement setup. Given that **B** and **I** are fixed in the same directions, this observation suggests that the OMR originates from the electrical transport along the side edges of the InAs thin film, where the SIS is broken by the opposite polarities, as discussed in the next paragraph. This argument is further supported by the disappearance of the OMR in our two-terminal resistance measured between electrodes 1 and 4 ($R_{14}$), where the positive and negative OMR components from the two side edges of the InAs thin film exactly cancel (see Fig. 2c and Supplementary Note 4). We note that, however, a large OMR was observed when we measured the resistance only along one edge by the two-terminal method (see Supplementary Note 5).

Two types of edge transport are known to occur in InAs/GaSb bilayers. One involves a non-trivial quantum spin Hall edge state[16–18], which is formed at the edge of the InAs/GaSb interface when a

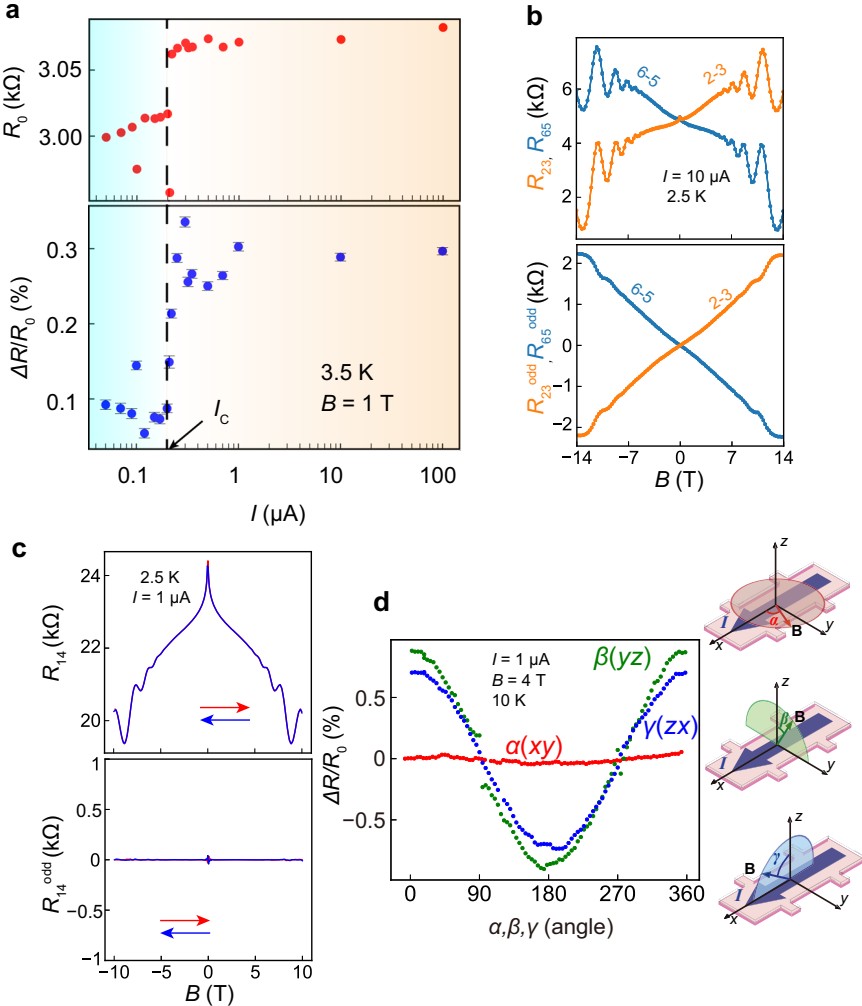

**Fig. 2 | Properties of the OMR in 1D InAs edge channels. a** Current $I$ dependence of the zero field resistance $R_0$ (upper panel) and the OMR magnitude $\Delta R/R_0$ (= $[(R_{23}(1\,\text{T}) - R_{23}(-1\,\text{T}))/2]/R_0$) (lower panel) measured in $R_{23}$ of sample B at 3.5 K. The $R_0$ and $\Delta R/R_0$ jump up at $I_C$ (= 200 nA) simultaneously. The error bars are obtained from the covariance of the linear fitting of the OMR. **b** Comparison of the $B$-dependences of $R_{23}$ and $R_{65}$ (upper panel) of sample A and their odd-function components (lower panel) measured with a fixed current of 10 μA at 2.5 K. $R_{23}$ and $R_{65}$, which are measured along the different 1D channels at the opposite edges, show opposite $B$ dependences. **c** MR curve of $R_{14}$ of sample A (upper panel) and its odd component (lower panel) measured with a fixed current of 1 μA at 2.5 K. The OMRs in the opposite 1D channels cancel each other out, leading to an almost zero odd component in $R_{14}$. **d** Angle dependences of the OMR magnitude $\Delta R/R_0$ of sample B, where $\Delta R = (R_{23}(4\,\text{T}) - R_{23}(-4\,\text{T}))/2$ and $R_0 = R_{23}(0\,\text{T})$. The red, green, and blue dots indicate the OMR magnitude in the $xy$, $yz$, and $zx$ rotations, respectively.

topological gap is opened due to the inverted band structure (the valence band top of GaSb is at higher energy than the conduction band bottom of InAs) and SOI. However, because this topological gap is very small (~4 meV), the non-trivial edge state cannot survive at high temperatures, which contradicts our observation of the OMR up to room temperature. The other involves a trivial edge state formed at the edge of the InAs layer due to the pinning of the Fermi level at the top and side vacuum surfaces, which is located as high as 0.1–0.3 eV above the conduction band bottom[19–24]. As a result, the conduction band potential of InAs is strongly bent downward at the surfaces, which we confirmed using Kelvin force microscopy measurements (See Methods and Supplementary Fig. 10). The effect is two-fold: First, the electron carriers accumulate more at the edges than in the center of the InAs film; thus, two 1D edge channels and one 2D transport channel coexist. This fact is confirmed by the transport measurements on devices with different sizes, which is discussed in Supplementary Note 6. Second, the SIS is broken at the side edges due to the resulting built-in electric field. Since we define the directions of **I** and **B** in our measurements as the $x$ and $z$ directions, respectively, as shown in Fig. 1a, the built-in electric field $\mathbf{E}_{\text{sur}}$ pushes the electron carriers outward along the $y$

direction. The directions of $\mathbf{E}_{\text{sur}}$ in the two edge channels are opposite, which explains the opposite signs of the OMRs in $R_{23}$ and $R_{65}$. As shown in Fig. 2d, the OMR almost disappears when we apply **B** parallel to the current **I** direction (the $x$-axis) or the $\mathbf{E}_{\text{sur}}$ direction (the $y$-axis) (see Supplementary Fig. 12). This indicates that OMR can only be induced when **B**, **I**, and $\mathbf{E}_{\text{sur}}$ are mutually orthogonal. This is also because the MPE from (Ga,Fe)Sb, which breaks the TRS in InAs, is only effectively induced by the $z$-component of the magnetization of (Ga,Fe)Sb[15].

## Control of SIS and TRS breaking via gate voltage

To examine our scenario, we apply an electrical gate voltage to individually tune the TRS and SIS breakings in the edge transport of InAs and evaluate their impacts on the OMR. We fabricated two field-effect transistor devices, D1 and D2; one (D1) has a single gate electrode, G, that controls the whole InAs Hall-bar (Fig. 3a), and the other (D2) has two separate gate electrodes, $G_1$ and $G_2$, that control the conduction of each edge independently (Fig. 3b). In device D1, a negative (positive) voltage applied to G push the electron wavefunctions in InAs towards the (Ga,Fe)Sb (top surface) side, which effectively enhances (suppresses) the MPE[15]. As shown in Fig. 3c, in device D1, when applying a

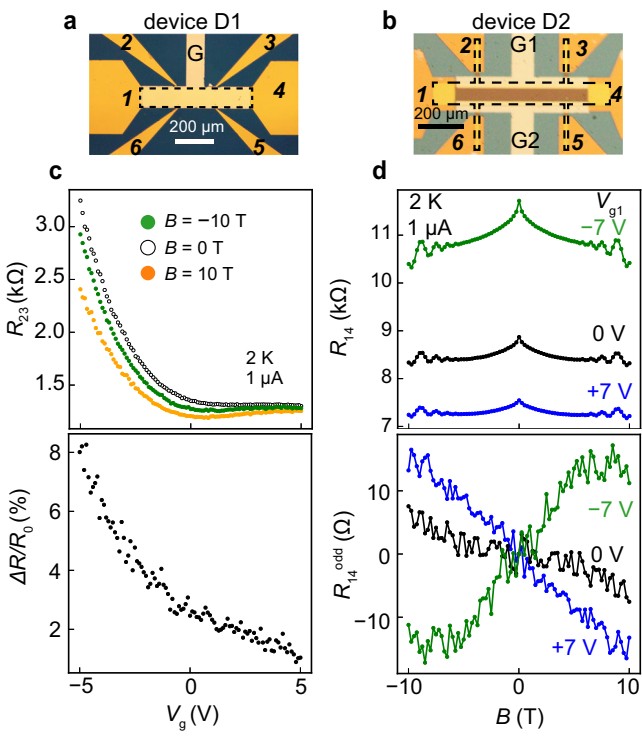

**Fig. 3 | Electrical gating of the OMR. a, b** Optical microscopy images of the Hall-bar field-effect transistor (FET) device D1 with a top-gate G and D2 with separated gates $G_1$ and $G_2$, respectively. These two devices are made from sample B. The dashed line indicates the outline of the Hall bars. The light and dark yellow parts are the Au pads of the gate and Hall-bar electrodes, respectively. **c** Gate voltage $V_g$ dependence of $R_{23}$ at various $B$ of −10 T (green), 0 T (white), 10 T (orange) (upper panel), and that of the odd component $\Delta R/R_0$ (lower panel), where $\Delta R = (R_{23}(-10\,\text{T}) - R_{23}(+10\,\text{T}))/2$, and $R_0 = R_{23}(0\,\text{T})$, measured at 2 K on device D1. These measurements are conducted with a fixed current of 1 μA at 2 K. **d** Magnetoresistance (MR) results of (top panel) and the odd components (bottom panel), measured at 2 K on device D1. $R_{14}$ is the resistance measured between terminals 1 and 4 (two-terminal measurement). The MR results at $V_{g1} = -7$, 0, +7 V are shown in green, black, and blue lines, respectively.

negative gate voltage $V_g$ from 0 to −5 V on G, with which the MPE is enhanced, the OMR intensity strongly increases by more than three-folds (2.5 to 8%, respectively). Meanwhile, when applying a positive $V_g$ from 0 to 5 V on G, which effectively suppresses the MPE, the OMR intensity decreases and almost vanishes at $V_g = -5$ V. These results clearly demonstrate the important role of TRS breaking in inducing the OMR. This fact is also confirmed by the small OMR magnitude (=1.8% at 14 T) in an InAs/GaSb reference sample, where there is no FM coupling, as shown in Supplementary Fig. 13. On the other hand, in device D2, by applying a voltage in one of these two gates (for example, G1), we modulate the band profile in one edge of the InAs layer (the edge along terminals 2 and 3). This enhances the OMR in one edge more than another and results in an appearance of OMR even in the magnetoresistance measured between terminals 1 and 4. Figure 3d shows the magnetoresistance characteristics measured between terminals 1 and 4 when we applied $V_{g1} = 7$ and −7 V on G1. One can see that a negative (positive) OMR is induced at $V_{g1} = 7$ V (−7 V), as expected. This can be understood because a positive (negative) $V_{g1}$ enhances (suppresses) the $\mathbf{E}_{sur}$ of the right edge relative to that of the left edge. Therefore, the important role of SIS breaking at the edge channels is clearly demonstrated by these results.

## Theoretical analysis

Finally, we discuss the theoretical model to explain the OMR in InAs/(Ga,Fe)Sb. If we temporarily neglect the MPE from the (Ga,Fe)Sb layer,

the Hamiltonian of the 1D edge channel of InAs can be described as

$$H_{1D}(k_x) = \frac{\hbar^2 k_x^2}{2m^*}\sigma_0 + (\Lambda_{\text{side}}k_x + \Delta_z)\sigma_z + \Lambda_{\text{top}}k_x\sigma_y \quad (1)$$

where $k_x$ is the wavenumber along the $x$-direction, $m^*$ is the effective mass of electrons, $\Lambda_{\text{top(side)}}$ $(=\hbar\lambda_{\text{top(side)}})$ is the effective Rashba SOI due to the built-in potential at the top (side edge) surface, $\hbar$ is the Dirac's constant, $\Delta_z$ $(=g\mu_B B_z)$ is the Zeeman splitting due to an applied magnetic field along the $z$-axis $(B_z)$, $\sigma_i$ $(i = x, y, z)$ are the elements of the Pauli matrix that acts on the electron spin degree of freedom, and $\sigma_0$ is the identity matrix. The energy dispersion from Eq. (1) can be described as

$$E_s = \frac{\hbar^2 k_x^2}{2m^*} + s\sqrt{(\Lambda_{\text{side}}k_x + \Delta_z)^2 + (\Lambda_{\text{top}}k_x)^2} \quad (2)$$

where $s = +/-$ denotes the upper and lower bands $E_+$ and $E_-$, as depicted in Fig. 4a, respectively. Here we define the energy band bottom of $E_-$ as $E = 0$. It is important to note that due to the Rashba SOI ($\Lambda_{\text{top}}$ and $\Lambda_{\text{side}}$), the spin components $\sigma_y$ and $\sigma_z$ are locked to the momentum $k_x$ in opposite directions between the bands $E_+$ and $E_-$. Thus the + and − subscripts also indicate the difference in "chirality" of these bands, which are shown as green and pink lines, respectively, in the right-side graph of Fig. 4a. We solve Boltzmann's equations and obtain the electrical conductivity $\sigma_{xx}$ by summing the conductivities of all the bands that cross the Fermi level ($E_F$) (see Supplementary Note 7),

$$\sigma_{xx} \simeq \frac{e^2}{h}\sum_s \tau_s \int dE_s \sqrt{1 + \frac{2E_s}{m^*\lambda_{\text{side}}^2}}\left(1 - s\frac{|\lambda_{\text{side}}|}{\lambda_{\text{side}}}\frac{\Delta_z}{m^*\lambda_{\text{side}}^2}\right)\delta(E_s - E_F) \quad (3)$$

where $e$ is the elementary charge, $\tau_s$ is the relaxation time, $h$ is Planck's constant, and $E_F$ is the Fermi energy. Reflecting the breaking of the SIS at the side surface edges, we assume $\Lambda_{\text{top}} \ll \Lambda_{\text{side}}$, which indicates that the electric field at the side edges is much larger than that at the top surface[25]. From Eq. (3) and Fig. 4a, the odd-order $B_z$-dependent conductivity can be non-zero in the case that $E_F$ crosses only the lower band shown in region (II) of Fig. 4a. However, this case is unlikely because the gap $\Delta_g(B)$ is only 24 and 44 meV at $B = 0$ and 14 T, respectively, obtained by using the parameters of an InAs nanowire of $m^*/m_0 = 0.08$[26], $g = 18$[27], $m^*\lambda_{\text{side}}^2 = 0.45$ meV[28], and $m^*\lambda_{\text{top}}^2 = 0.027$ meV[28]. Due to the Fermi level pinning at the edge surface, $E_F$ in the edge channel lies in the region (I) of Fig. 4a, where the odd-order $B_z$-dependent conductivities from the upper and lower bands cancel each other out, and thus, no OMR should be expected.

However, the OMR can be induced if the relaxation times in the $E_+$ and $E_-$ bands are different ($\tau_+ \neq \tau_-$), which results from the MPE and the Rashba SOI in the 2D and 1D channels of InAs as explained in the following. Our analysis of the transport data (see Supplementary Notes 2, 6) indicates that the MPE mainly affects the 2D channel, inducing a splitting energy gap $\Delta_{2D}$ between 2D bands of opposite $\sigma_z$ components (indicated by purple arrows in Fig. 4b). Therefore, the MPE affects the 1D channel only indirectly via electron scattering between the 1D and 2D channels. Considering that $\sigma_y$ is locked to $k_x$ because of $\Lambda_{\text{top}}$ in both the 1D and 2D channels, the lower and upper bands in each channel (1D and 2D) have different chiralities, as indicated by the green and pink colors in Fig. 4b. The relaxation time of $E_+$ and $E_-$ ($\tau_+$ and $\tau_-$, respectively) in the 1D channel thus are mainly determined by scattering events between bands with the same chirality (blue and red arrows in Fig. 4b). The difference in the density of states (DOS) at the Fermi level of the two 2D bands (pink and green bands in Fig. 4b) then leads to the asymmetric scattering between $E_+$

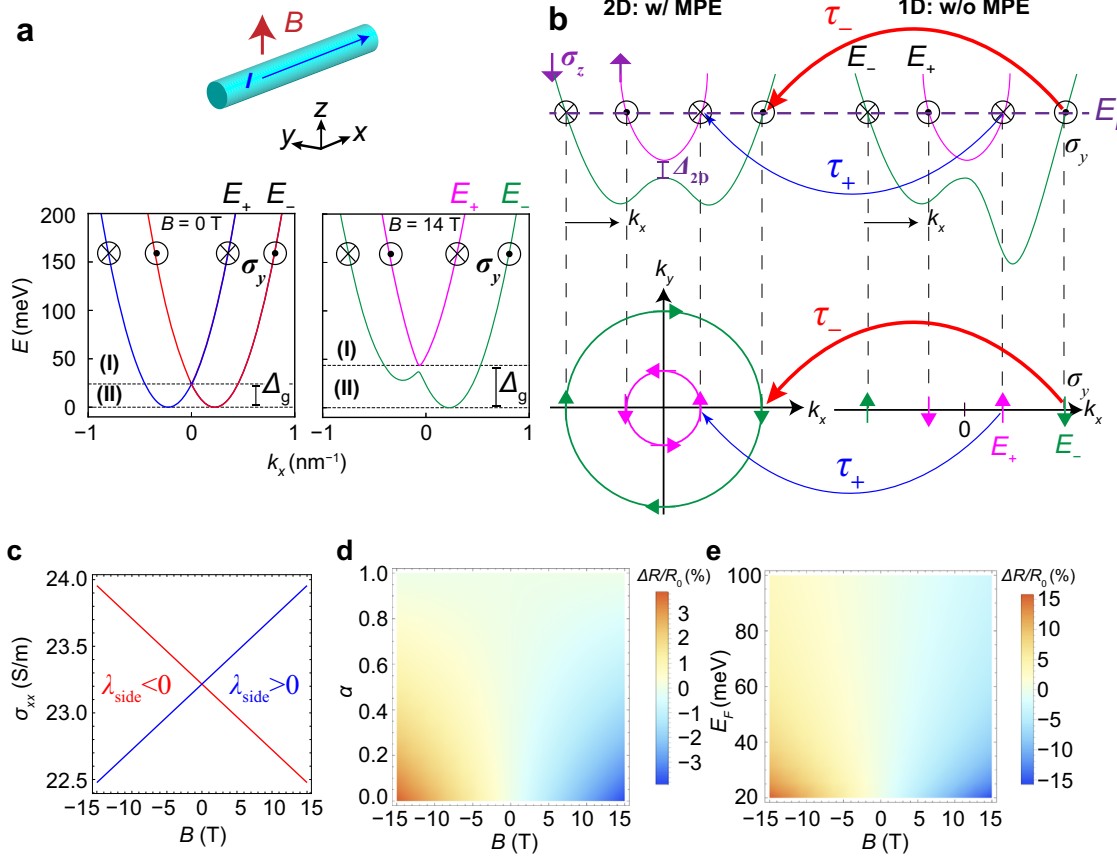

**Fig. 4 | Theoretical calculation using Boltzmann's equation in the 1D Rashba system with an MPE. a** Energy band dispersions ($E_+$, $E_-$) of the 1D Rashba system under a magnetic field $B$ (0 T and 14 T, shown in the left and right panels, respectively) applied parallel to the $z$-direction, calculated by Eq. (2). Here, we set $m^*/m_0 = 0.08^{26}$, $\Delta_z/B_z = 0.52$ meV/T for $g = 18^{27}$, $m^*\lambda^2_{side} = 0.45$ meV[28], and $m^*\lambda^2_{top} = 0.027$ meV[28]. In the case of $B = 14$ T, $E_+$ and $E_-$ can be labeled by chirality (pink and green lines, respectively) which is determined by the SIS breaking in the $z$-direction. Two different regions (I) and (II) can be observed, defined by whether the Fermi energy $E_F$ crosses only one or two dispersion branches $E_+$ and $E_-$. $\Delta_g$ is the energy gap between the minima of $E_+$ and $E_-$. **b** Schematic energy dispersion (upper) and its Fermi surface (lower) of the 1D (right) and 2D (left) channels. The purple dashed line indicates the Fermi level $E_F$. Due to the Rashba SOI in the $z$ direction in both channels, the $y$ spin component ($\sigma_y$) of electrons is locked to the momentum $k_x$ in opposite directions between the green and pink bands. Electron scattering between the 1D and 2D channels occurs mainly between bands with the same chirality, as indicated by the red and blue arrows. In the 2D channel, MPE opens the gap ($\Delta_{2D}$) between two bands with opposite $z$ spin components ($\sigma_z$). Different density of states between the two (pink and green) 2D bands leads to different relaxation times $\tau_+$ and $\tau_-$ in the 1D channel. **c** Calculated results of the OMR using Eq. (4) with $\alpha = 0.1$, $E_F = 100$ meV, $m^*\lambda^2_{side} = 0.45$ meV[28], and $\Delta_z/B_z = 0.52$ meV/T for $g = 18^{27}$. The sign of the Rashba parameter $\lambda_{side}$ determines the polarity of the OMR component in the 1D system. **d, e** OMR as functions of $\alpha$ (with $E_F = 100$ meV) and $E_F$ (with $\alpha = 0.1$), respectively. $\alpha$ (= $\tau_-/\tau_+$) represents the different relaxation times of electron carriers in the $E_-$ and $E_+$ states; $\alpha$ is small in the case of a strong MPE. A strong MPE and a small $\alpha$ lead to a large OMR.

and $E_-$ in the 1D edge channel, and different values of $\tau_+$ and $\tau_-$ (see Supplementary Note 8 and 9 for detailed discussions). Consequently, the linear-response conductivity $\sigma_{xx}$ is rewritten as

$$\sigma_{xx} \simeq \frac{e^2}{h}\, \tau_+ |\lambda_{side}| \sqrt{1 + \frac{2E_F}{m^*\lambda^2_{side}}} \left[1 + \alpha - (1-\alpha)\frac{|\lambda_{side}|}{\lambda_{side}}\frac{g\mu_B}{2E_F + m^*\lambda^2_{side}}B_z\right] \tag{4}$$

Here, we set the phenomenological parameter $\alpha$ as $\tau_- = \alpha\tau_+$ to express the different relaxation times of electron carriers in the $E_+$ and $E_-$ states. Under the influence of the strong MPE and chirality-dependent scattering at the interface ($\alpha \ll 1$), the linear $B_z$-dependent MR appears in the conductivity $\sigma_{xx}$ due to the contribution of the last term in the brackets of Eq. (4). Using $\alpha = 0.1$, $E_F = 100$ meV, $m^*\lambda^2_{side} = 0.45$ meV[28], and $\Delta_z/B_z = 0.52$ meV/T for $g = 18^{27}$, the OMR is clearly reproduced by Eq. (4), as shown in Fig. 4c. The different signs of $R_{23}^{odd}$ and $R_{65}^{odd}$ shown in Fig. 2b are explained by the different signs of the Rashba parameter $\lambda_{side}$ (blue and red lines) between the two side edges. The dependences of the OMR ratio $\Delta R/R_0$ on $\alpha$ and $E_F$ are shown in Fig. 4d and e, respectively. A large difference in the relaxation time of the spin

channels, which means a small $\alpha$, produces a large OMR ratio. This indicates the important role of MPE at the InAs/(Ga,Fe)Sb interface in inducing the large OMR. This conclusion is also supported by the fact that the OMR magnitude $\Delta R/R_0$, where $\Delta R = R_{23}^{odd}(10$ T) and $R_0 = R_{23}(0$ T), is enhanced with decreasing temperature $T$ as $\ln(1/T)$ (see the inset of Fig. 1d). This behavior is characteristic of the Kondo-effect-related transport coming from the spin-dependent scattering at the InAs/(Ga,Fe)Sb interface. Another important result is that a smaller $E_F$ leads to a larger OMR. If we set $E_F$ at approximately 24 meV, which is the same as $\Delta_g(0$ T), then Eq. (4) can reproduce the experimental value ($\Delta R/R_0 = 13.5\%$), as shown in Fig. 4e.

In conclusion, we found the giant odd-parity magnetoresistance in the 1D edge channels of the InAs/(Ga,Fe)Sb heterostructure, and demonstrated the ability to electrically turn on and off the effect using field-effect transistor structures. Our results highlight the abundance of new physics in solid-state systems when TRS and SIS are simultaneously broken, even in well-known materials such as InAs. The linear OMR presented in this work can be applied to magnetic field sensors, which provide a large dynamic range (0–10 T) owing to its linearity. This new type of sensor can work at room temperature, requires only

simple DC measurements for detection, and its sensitivity can be further enhanced by material engineering, such as optimizing the carrier concentration and SOI strength.

## Methods

### Sample preparation and characterization

We grew heterostructures consisting of InAs (thickness 15 nm)/ (Ga,Fe)Sb (15 nm, Fe 20%, $T_C$ > 300 K)/AlSb (300 nm)/AlAs (15 nm)/ GaAs (100 nm) on semi-insulating GaAs (001) substrates by molecular beam epitaxy (MBE). The growth temperature ($T_S$) was 550 °C for the GaAs and AlAs layers, 470 °C for the AlSb layer, 250 °C for the (Ga,Fe)Sb layer, and 235 °C for the InAs layer. We also grew a non-magnetic InAs/GaSb heterostructure as a reference, whose structure is the same as the sample mentioned above, except for the lack of Fe doping. The top two layers (InAs and GaSb) of this sample were grown at 470 °C, while the other layers were grown under the same conditions as the Fe-doped samples. The in situ reflection high energy electron diffraction (RHEED) patterns of InAs and (Ga,Fe)Sb are bright and streaky, indicating good crystal quality and a smooth surface (see Supplementary Fig. S2b in ref. [15]). In this paper, we used two different samples A and B of InAs/(Ga,Fe)Sb heterostructures with sheet carrier concentrations of $2.0 \times 10^{12}$ cm$^{-2}$ and $1.8 \times 10^{12}$ cm$^{-2}$, and electron mobilities of $9.4 \times 10^2$ cm$^2$/Vs and $1.9 \times 10^3$ cm$^2$/Vs, respectively. Also, the quantum mobility of sample A is estimated to be 2070 cm$^2$/Vs from the SdH oscillations (See Supplementary Fig. 14).

The mobility difference between samples A and B suggests that the static electric fields, which determine the confinement potential in the 1D and 2D channels of InAs, in the two samples are different. The confinement potential sensitively affects the strength of both the proximity magnetoresistance[15] and the Rashba SOI in the InAs channel, and consequently yields different OMR in these samples A and B. We note that the different confinement potentials may originate from different surface pinning effects at the top and side surfaces of the devices, which depend on the detailed conditions during device fabrication.

### Fabrication process of the Hall bar devices and transport measurement

The samples were patterned into $100 \times 600$ μm$^2$ Hall bars using standard photolithography and Ar ion milling down to the AlSb buffer layer. The etched surface was passivated by depositing a thin SiO$_2$ layer. Then electrodes were formed by electron-beam evaporation and lift-off of Au (50 nm)/Cr (5 nm) films. Figure 1b shows an optical microscopy image of the Hall bar device examined in Figs. 1, 2, and 3c, d. For the field-effect transistor (FET) devices in Fig. 3c, d, we deposited a 50-nm-thick Al$_2$O$_3$ layer as a gate insulator by atomic layer deposition. Figure 3a, b show optical microscopy images of the Hall bar FET device examined in Fig. 3c (D1) and d (D2), respectively. Magnetotransport measurements were conducted using a Quantum Design physical property measurement system (PPMS) by a standard 4-terminal method, except for $R_{14}$, which was measured by a two-terminal method. We use a DC current for $I$ > 1μA, and an AC current with a lock-in amplifier (lock-in frequency is 5261 Hz) for lower $I$.

Supplementary Fig. 10a shows the atomic force microscope (AFM) images of the InAs/(Ga,Fe)Sb sample before (left) and after (right) etching using Ar ion milling, similar to the fabrication process of the field-effect transistors in our work. The root mean square (RMS) of the roughness is only 0.4 nm, which is less than two monolayers of InAs and remains almost unchanged after the etching. Also, no apparent etch pit is observed in these AFM images. This result indicates that our films are homogeneous and smooth, which is not affected by the device fabrication process.

### Work function measurements by Kelvin probe force microscopy (KFM)

We investigated distribution of the surface potential on the InAs/ (Ga,Fe)Sb by KFM in vacuum conditions (-10$^{-5}$ Pa) at room temperature. In KFM, an AC bias at frequency $f$ (=1 kHz in our case) and a DC bias are applied between the tip and the sample under noncontact operation in atomic force microscopy (AFM) (see Supplementary Fig. 10b and c in Supplementary Information). When the tip approaches the sample surface in the $z$-direction, the electric bias induces an electrostatic force $F$ expressed as

$$
\begin{aligned}
F = &\frac{1}{2}\frac{dC}{dz}\left(V_{dc} - \frac{\Delta\phi}{e} + V_{ac}\sin 2\pi ft\right)^2 \\
= &\frac{1}{2}\frac{dC}{dz}\left(V_{dc} - \frac{\Delta\phi}{e}\right)^2 + \frac{1}{4}\frac{dC}{dz}V_{ac}^2 + \frac{dC}{dz}\left(V_{dc} - \frac{\Delta\phi}{e}\right)V_{ac}\sin 2\pi ft \\
&- \frac{1}{4}\frac{dC}{dz}V_{ac}^2\cos 4\pi ft,
\end{aligned}
\tag{5}
$$

where $C$ is the capacitance between the tip and the sample, $V_{dc}$ is the DC bias voltage, $V_{ac}$ is the AC voltage magnitude, and $\triangle\phi$ is the work function difference between the tip and the sample. Similar to AFM measurements, the force $F$ is deduced from the shift of the cantilever oscillation frequency. $V_{dc}$ is adjusted using a feedback control so that the $f$-frequency component in $F$, which is measured using a lock-in amplifier, is nullified. Then $V_{dc}$ gives the value of $\Delta\phi/e$ according to Eq. (5). Therefore, we can obtain $\triangle\phi$ and consequently, the work function distribution on the sample.

We note that the potential profile at the topmost InAs surface detected by KFM might be different from the one at several-atomic-layer depth below the surface. This is because of a screening effect from a large amount of charged surface states (top and side surfaces), which are common at InAs surfaces. Thus, the potential profile change along the $y$ direction measured by the KFM tip is much milder than the real confinement potential at the edge of the InAs channel[29]. As a result, the potential profile at the top surface shown in Supplementary Fig. 10c might largely exaggerate the width of the triangular potential at the bulk InAs side surface, which should be much less than 2 μm. Therefore, we consider that the static electric field in the $y$ and $z$ directions should have the same order of magnitude.

## Data availability

All data generated in this study are provided in Source Data[30][https:// doi.org/10.5281/zenodo.7141370].

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

## Acknowledgements

This work was partly supported by Grants-in-Aid for Scientific Research (Nos. 16H02095, 17H04922, 18H05345, 20H05650, and 20K15163), the CREST Program (JPMJCR1777), and PRESTO Program (JPMJPR19LB) of the Japan Science and Technology Agency, and the Spintronics Research Network of Japan (Spin-RNJ). A part of this work was conducted at the Advanced Characterization Nanotechnology Platform of the University of Tokyo, supported by the "Nanotechnology Platform" of the Ministry of Education, Culture, Sports, Science and Technology (MEXT), Japan. K.T. acknowledges the support from the Japan Society for the Promotion of Science (JSPS) Fellowships for Young Scientists. K.T. and R.F. acknowledge the Material Education program for future leaders in Research, Industry, and Technology (MERIT). L.D.A. acknowledges the support from the UTEC-UTokyo FSI research granting program and the Murata Science Foundation.

## Author contributions

K.T. and L.D.A. designed the experiments and grew the samples. K.T. fabricated devices, performed sample characterizations, and transport measurements. R.F. and T.T. conducted KFM measurements. H.S. conducted AFM measurements. K.T., L.D.A., T.C., and M.T. discussed the mechanism. T.C. performed theoretical calculations. K.T., L.D.A., and M.T. wrote the manuscript. L.D.A. and M.T. supervised the study.

## Competing interests

The authors declare no competing interests.

## Additional information

**Correspondence and requests** for materials should be addressed to Le Duc Anh or Masaaki Tanaka.

