## [Peer Review File · Nature Communications]

Reviewers' Comments:

Reviewer #1:

Remarks to the Author:

The authors have made several factual mistakes in their introduction and literature survey, which also led to their misunderstanding of the mechanisms of odd-parity linear MR. Their misunderstanding has significantly affected their experimental plan, credibility of data analysis, and theoretical discussion.

First of all, Onsager's reciprocal relationship represents a thermodynamic principle, so it is always correct as long as all entities breaking time reversal symmetry (such as B and M) are accounted (reversed with time reversal). This principle concerns only the time reversal symmetry, but not inversion symmetry; inversion symmetry is a point group symmetry and has nothing to do with thermodynamics. Spin-orbit coupling does not break the time reversal symmetry. So the authors' statements in the abstract are inaccurate. Their statements on lines 14-16 of page 2 are even wrong; as a matter of fact, all previous systems that demonstrated odd-parity LMR, as listed in their Supplementary Table 1, preserves inversion symmetry in lattice (such as SmCo₅ and Eu₂Ir₂O₇).

The authors' literature survey is far from accurate and complete. First of all, although Ref. 8 stated in their paper the cause of odd-parity LMR in Eu₂Ir₂O₇ is from the all-in-all-out antiferromagnetism, they were wrong. The measurement geometry in Ref. 8 is unable for the all-in-all-out AFM in a pyrochlore structure to generate odd-parity MR, as this has been theoretically proven by Xiao et al. PRB 101, 201410R (2020) based on rank-3 pseudotensor. Ref. 7 actually discussed two mechanisms of odd-parity MR in SmCo₅. The first should have the same origin as Ref. 6 that bulk FM (but not non-uniformity) can introduce odd-parity MR based on the Zeeman effect. The second mechanism is from the anomalous Hall effect such as Xiao et al. PRB (2020) discussed. Ref. 7 also discussed the all-in-all-out system Cd₂Os₂O₇, which hosts the same type of antiferromagnetism as Eu₂Ir₂O₇, and the odd-parity LMR was proven to be from domain walls but not all-in-all-out AFM. So the summary in the Supplementary Table 1 is very inaccurate with regard to the existing known mechanisms and systems.

This misunderstanding of the literature has significantly impacted the authors' ability to establish their experimental case. While the influence of ferromagnetism to odd-parity MR is well known by now, here in the presence of a FM substrate, the question always remains whether this is from similar mechanisms established previously. It's the authors' burden to prove against all existing mechanisms if they want to establish their results for a new mechanism of odd-parity MR. So far the gate voltage control is not convincing with regard to removing the effect of FM substrate. As stated before, spin-orbit coupling alone could not introduce odd-parity MR, and if the author is going to claim such, they need to establish it in the absence of an FM substrate. Because of the authors' misunderstanding of Onsager's reciprocal relationship, their theoretical model is largely technical and lacks a clear picture of the odd-parity physics.

The phenomena these authors observed might have already been accounted in the literature. In a recent paper by Zyuzin PRB 104, L140407(2021), it was discussed in the presence of both magnetization and a chiral state, which has the inversion symmetry broken, the finite amount of spin-orbit coupling seems can be coupled with the ferromagnetism for the odd-parity MR. It is clear that the presence of ferromagnetism is always necessary in this mechanism of linear MR, and SOC seems serves to enhance the linear MR effect due to its influence of the band structure. The SOC enhanced odd-parity MR has been experimentally discussed in a magnetic Weyl semimetal (Jiang et al., PRL 126, 236601 (2021)). So I think this mechanism is reasonably well established both experimentally and theoretically. The authors' experimental plan of using magnetic proximity effect actually makes the physics more ambiguous.

So with the current work not well written, and missing clear experimental logic, I believe it is not suitable for Nat Comm.

Reviewer #2:

Remarks to the Author:

The manuscript, titled by "Giant gate-controlled odd parity magnetoresistance in one-dimensional channels with a magnetic proximity effect", reports an interesting phenomenon in InAs/GaFeSb heterostructure. The authors observed a giant odd parity magnetoresistance (OMR) as large as 27% in an InAs quantum well, which may be induced by a proximity effect from underlying ferromagnetic semiconductor GaFeSb. They also show that the OMR can be controlled by electrical gating either with a top-gate or separated edge gates. In addition, a theoretical analysis is proposed to explain the observed OMR.

The experimental results are very interesting and it would bring a lot attraction in the field of spintronics. However, both experiments and theoretical analysis have lots of issues to be resolved. Regarding experiments, it seems that the OMR can be any values in the devices the authors measured, from 27% at $\pm 10T$, to 0.6% at $\pm 1T$. In particular, Fig. 2, the data in 2a and 2d are not consistent with Data in 2b. Could the authors explain what makes the OMR value varies so much in one heterostructure?

On the other hand, if the authors think both 1D edge and 2D transport channels coexist, what is the current distribution ratio between them? If the edge channel is dominated in the transport, how is the OMR depending on the width of the Hall bar? Does the top gate control such ratio? What is the Hall bar orientation respect to the film crystallographic direction?

Does such OMR depend on the Hall bar orientation?

The oscillation frequency of SdH represents what channel of conductivity? What is the electron density, Fermi energy and mobility obtained from SdH oscillation?

The surface pinning also happens on InAs top surface, why is the edge (2 micron from KFM) is so different from top surface of InAs? Is this the nano fabrication effect?

What is the real morphology profile of the edge of the InAs/GaFeSb Hall bar?

Regarding theoretical analysis, two key issues the authors should clarify. 1. The structure inversion symmetry is broken at the interface between InAs and GaFeSb. What effect is such breaking on the OMR analysis?

2. In Fig. 4a, why do both the side Rashba effect and magnetic field (Zeeman effect) make electron spin degenerate in z direction?

Reviewer #3:

Remarks to the Author:

Electron and spin transport in InAs/GaSb (a quantum spin Hall system) has attracted large attention in recent years. In this work, a more interesting system InAs/GaFeSb (ferromagnetic) was investigated. In this system, a very interesting and rare OMR was observed. The authors performed well-designed experiments. The model is also reasonable. I recommend to publish this paper in Nature Communications after the authors tackle several minor points.

1. The Hall measurement results of the system should be very interesting in. I believe that every reader of this paper wants to see the Hall measurement results, which can provide a lot of important information to understand this interesting OMR effect and improve the quality of the paper. Could the authors show the results somewhere in supplementary?

2. The band structure shown in Figure 4a induces spin momentum locking in the edge transport. Could spin momentum locking has some relationship with this interesting OMR?

3. How do you define the spin-up and spin-down electrons? As the spin is also related to momentum direction, can we define the spin-up and spin-down based on the magnetization direction?

Response Letter

We would like to thank the reviewers for their reports on our manuscript submitted for publication in Nature Communications (# NCOMMS-21-39056). We have carefully revised our manuscript in accordance with the reviewers' comments. In the following, we show point-by-point response to each comment and how we revised our main manuscript and Supplementary Information. The revised parts are indicated by red letters.

Reviewer #1:

Comment 1-1

The authors have made several factual mistakes in their introduction and literature survey, which also led to their misunderstanding of the mechanisms of odd-parity linear MR. Their misunderstanding has significantly affected their experimental plan, credibility of data analysis, and theoretical discussion.

First of all, Onsager's reciprocal relationship represents a thermodynamic principle, so it is always correct as long as all entities breaking time reversal symmetry (such as B and M) are accounted (reversed with time reversal). This principle concerns only the time reversal symmetry, but not inversion symmetry; inversion symmetry is a point group symmetry and has nothing to do with thermodynamics. Spin-orbit coupling does not break the time reversal symmetry. So the authors' statements in the abstract are inaccurate. Their statements on lines 14-16 of page 2 are even wrong; as a matter of fact, all previous systems that demonstrated odd-parity LMR, as listed in their Supplementary Table 1, preserves inversion symmetry in lattice (such as SmCo5 and Eu2Ir2O7).

Our response:

We thank the referee for his/her valuable comments. Based on these comments, we have carefully revisited the literature and revised the confusing or incorrect information in the previous manuscript.

First, we would like to correct some descriptions in our previous manuscript, which may have been confusing and led to misunderstandings. **We did not claim that spatial inversion symmetry (SIS) breaking and SOI alone can violate Onsager's reciprocal theorem and yield odd-parity linear magnetoresistance (OMR). We totally agree with the reviewer that, in principle, breaking the time reversal symmetry (TRS) is always necessary for observing OMR.** As a matter of fact, the large OMR observed in our InAs/(Ga,Fe)Sb bilayer heterostructures results from the breaking of TRS by the magnetic proximity effect coming from the magnetization of the ferromagnetic semiconductor (Ga,Fe)Sb layer. We will show experimental evidence and detailed discussion on the role of TRS breaking in our samples in the response to comment 1-3 and 1-5 below. However, in the experimental cases where TRS was broken and OMR was reported so far, SOI also plays an important supporting role to *enhance* the linear OMR effect through altering the band structure of the material, as also mentioned by the reviewer. This is what we meant in the previous manuscript, where we described that both TRS breaking and strong SOI are important to experimentally observe OMR in a material.

Second, we acknowledge that SIS breaking is not necessarily required for the appearance of OMR (on this point, our description was actually misleading in our previous

manuscript). As the reviewer has correctly pointed out, in many materials where OMR was observed previously (SmCo₅ and Eu₂Ir₂O₇), the SIS is preserved although strong SOI exists. We have revised these parts in the abstract, introductory paragraph, and Supplementary Table 1. We really thank the reviewer for this important comment.

In our InAs/(Ga,Fe)Sb bilayer heterostructures, however, SIS breaking plays an important role of yielding large Rashba SOI in the InAs edge channels. When TRS is broken in the InAs layer by the magnetic proximity effect from the underlying ferromagnetic (Ga,Fe)Sb layer, the strong Rashba SOI induced by the broken SIS enhances the OMR to unprecedentedly large values (~13.5% at 0 – 10 T, ~27% at ±10 T). The effects of both TRS breaking and Rashba SOI are adequately taken into account in our theoretical model, as can be seen in the Hamiltonian of the one-dimensional (1D) InAs edge channel given below (eq.(1) in the previous/revised main manuscript).

$$\hat{H}_{1D}(k_x) = \frac{\hbar^2 k_x^2}{2m^*} \sigma_0 + (\Lambda_{\text{side}} k_x + \Delta_z) \sigma_z + \Lambda_{\text{top}} k_x \sigma_y \quad (\text{R1})$$

where k_x is the wavenumber along the x direction (InAs edge channel direction), m^* is the effective mass of electrons, \hbar is Dirac's constant, σ_i ($i = x, y, z$) are the elements of the Pauli spin matrix that act on the electron spin degree of freedom, and σ_0 is the identity matrix. Here, Δ_z is the Zeeman splitting due to an applied magnetic field along the z -axis (B_z), which breaks the TRS, and $\Lambda_{\text{top(side)}} (= \hbar\lambda_{\text{top(side)}})$ is the effective Rashba SOI due to the built-in potential that breaks the SIS at the top (side edge) surface of the InAs edge channel. Again, we emphasize that the TRS breaking term $\Delta_z \sigma_z$ is necessary to violate the Onsager theorem and leads to the observation of very large OMR in our samples.

➤ **Corresponding revised parts in the manuscript:**

- In order to correct the misleading statements, we revised following parts in the main manuscript.
- ✧ lines 27-28 page 1 in the abstract:
 - (Old) “..., which usually involve time reversal symmetry (TRS) broken by ferromagnetism and spatial inversion symmetry (SIS) broken by strong spin orbit interactions, ...”
 - (New) “..., which involve time reversal symmetry (TRS) broken by ferromagnetism, ...”
- ✧ lines 1-3 page 2 in the main manuscript:
 - (Old) “... simultaneous breaking of both SIS in the one-dimensional (1D) InAs edge channels and the TRS by the magnetic proximity effect (MPE) is the origin of this giant OMR”
 - (New) “... simultaneous breaking of both the TRS by the magnetic proximity effect (MPE) and spatial inversion symmetry (SIS) in the one-dimensional (1D) InAs edge channels is the origin of this giant OMR.”
- ✧ line 14 page 2 in the main manuscript:
 - (Old) “The odd-parity MR (OMR) in a linear response regime is a novel phenomenon only observed in systems where both TRS and spatial inversion symmetry (SIS) are simultaneously broken.”

(New) “*The odd-parity MR (OMR) in a linear response regime has been observed in systems where TRS is violated.*”

- We added the suffix of “ x ” to k (wavevector) in eq. (1) in order to clarify the direction of the 1D channel like eq. (R1).

Comment 1-2

The authors’ literature survey is far from accurate and complete. First of all, although Ref. 8 stated in their paper the cause of odd-parity LMR in Eu₂Ir₂O₇ is from the all-in-all-out antiferromagnetism, they were wrong. The measurement geometry in Ref. 8 is unable for the all-in-all-out AFM in a pyrochlore structure to generate odd-parity MR, as this has been theoretically proven by Xiao et al. PRB 101, 201410R (2020) based on rank-3 pseudotensor. Ref. 7 actually discussed two mechanisms of odd-parity MR in SmCo₅. The first should have the same origin as Ref. 6 that bulk FM (but not non-uniformity) can introduce odd-parity MR based on the Zeeman effect. The second mechanism is from the anomalous Hall effect such as Xiao et al. PRB (2020) discussed. Ref. 7 also discussed the all-in-all-out system Cd₂Os₂O₇, which hosts the same type of antiferromagnetism as Eu₂Ir₂O₇, and the odd-parity LMR was proven to be from domain walls but not all-in-all-out AFM. So the summary in the Supplementary Table 1 is very inaccurate with regard to the existing known mechanisms and systems.

Our response:

We thank the reviewer for informing the recent theoretical discussions on the possible origins of OMR in previous studies. To provide readers with wider information, we have revised the details in Supplementary Table 1, as shown in Table R1 below.

Table R1. Comparison of OMR observed in previous reports and our work. The maximum OMR magnitude $\Delta R/R_0$ ($= [(R(B)-R(-B))/2]/R(0\text{ T})$) normalized by R_0 ($= R(0\text{ T})$) were obtained under magnetic field B at temperature T .

Material	$\Delta R/R_0$ (%)	B (T)	T (K)	Observable under B // M	Proposed origin	Refs.
SmCo ₅	1.3×10^{-2}	0.015	room temp.	No	non-uniform distribution of the magnetization	R1
SmCo ₅	4.6×10^{-2}	0.5	300	No	Zeeman splitting/ anomalous Hall effect	R2, R3
Gd ₂ Os ₂ O ₇	5.0×10^{-2}	2	195	No	magnetic domain walls	R3
Eu ₂ Ir ₂ O ₇ (theory)	-	-	-	No	Berry curvature, magnetic moment, and shift vector	R2
Eu ₂ Ir ₂ O ₇ (experiment)	0.44	9	2	No	magnetic texture	R4
Fe ₃ GeTe ₂ / graphite/ Fe ₃ GeTe ₂	1.1	0.01	50	No	interfacial SOI of Fe ₃ GeTe ₂ as a topological nodal line	R5
InAs/ (Ga,Fe)Sb	13.5 5	10 10	2 300	Yes	Rashba SOI at the edge of InAs and magnetic proximity effect	Our work

➤ **Corresponding revised parts in the manuscript:**

We revised Supplementary Table 1 as shown in Table R1. (Revised points are shown as red letters) For broad interest of readers, we show all the proposed origins of OMR in each work.

- ✧ For the case of SmCo₅ from Ref R3 Wang, Y. et al. Nat. Commun. **11**, 216 (2020), we revised the origin of OMR, adding “Zeeman splitting/ anomalous Hall effect” in Supplementary Table 1.

- ✧ For the case of $\text{Eu}_2\text{Ir}_2\text{O}_7$, we added the origin of OMR as “Berry curvature, magnetic moment, and shift vector” based on C. Xiao et al. *Phys. Rev. B* **101**, 201410R (2020).
- ✧ We added the case of $\text{Gd}_2\text{Os}_2\text{O}_7$ from Y. Wang, et al. *Nat. Commun.* **11**, 216 (2020) (Ref. 7 in the main manuscript).
- ✧ We added the fifth column (**Observable under $\mathbf{B} // \mathbf{M}$**) of Table R1, which indicates whether the OMR is observable or not when \mathbf{M} is parallelly aligned with \mathbf{B} . The detailed discussions on this feature are given in our response to comment 1-3 below.

Comment 1-3

This misunderstanding of the literature has significantly impacted the authors’ ability to establish their experimental case. While the influence of ferromagnetism to odd-parity MR is well known by now, here in the presence of a FM substrate, the question always remains whether this is from similar mechanisms established previously. It’s the authors’ burden to prove against all existing mechanisms if they want to establish their results for a new mechanism of odd-parity MR.

Our response

While the influence of ferromagnetism on the OMR is common in all the previous reports and ours, there is one fundamental difference between the OMR observed in our InAs/(Ga,Fe)Sb bilayer heterostructures and the others. As shown in Table R1, in the previous reports, the linear OMR only occurred when the magnetic field \mathbf{B} and the magnetization \mathbf{M} are *separately* changed (thus they are not always parallel, $\mathbf{B} \nparallel \mathbf{M}$). This is because, although each \mathbf{B} and \mathbf{M} can break TRS, simultaneous reversal of both \mathbf{B} and \mathbf{M} preserves the TRS. Therefore, when \mathbf{B} and \mathbf{M} are completely aligned ($\mathbf{B} // \mathbf{M}$, which is the case when \mathbf{B} is large), the Onsager’s reciprocal theorem requires $\sigma_{xx}(\mathbf{B}, \mathbf{M}) = \sigma_{xx}(-\mathbf{B}, -\mathbf{M})$ and no OMR is allowed (σ_{xx} is longitudinal conductivity), as discussed in C. Xiao et al. *Phys. Rev. B* **101**, 201410R (2020). However, when $\mathbf{B} \nparallel \mathbf{M}$, the TRS is broken by reversing either \mathbf{B} or \mathbf{M} alone, which relaxes the Onsager’s theorem requirement and allows OMR to occur. For example, the OMR in $\text{SmCo}_5^{\text{R1, R3}}$ is observed only when \mathbf{B} is smaller than the coercivity of \mathbf{M} (~ 2 T). Thus, previously reported OMR phenomena were only realized in small magnetic field regions.

In contrast, the OMR in our InAs/(Ga,Fe)Sb system is fundamentally different. In our case, the OMR is present and linearly proportional to the magnetic field \mathbf{B} in the whole range of \mathbf{B} up to 10 T, which is much larger than the coercivity of (Ga,Fe)Sb (~ 50 mT). It is obvious that the magnetization \mathbf{M} of (Ga,Fe)Sb should closely follow \mathbf{B} in most of the magnetic field range (*i.e.* $\mathbf{B} // \mathbf{M}$). Therefore, the observation of OMR in our case is striking, because the TRS should be preserved when the both \mathbf{B} and \mathbf{M} are simultaneously reversed and always parallel, as mentioned above. Thus, this suggests that the OMR in our InAs/(Ga,Fe)Sb system should have a different origin and cannot be explained by the same theoretical framework of the previous reports.

At the present stage, we do not have a rigorous theoretical explanation for the large OMR observed in our InAs/(Ga,Fe)Sb system. However, our idea is that there should be some other factors that break the TRS, which only weakly depends on the external magnetic field \mathbf{B} . Here we show a possible mechanism to explain our OMR results. Figure R1a illustrates the band dispersions of the 1D edge channel of InAs, where there are two branches of energy dispersion E_+ and E_- . These dispersions are the results of the Rashba spin orbit

interaction (SOI) at the top (z direction) and side (y direction) surface of the InAs edge (see Fig.4a in the main manuscript or Fig. R1a below). The eigenvalues of these E_+ and E_- branches were obtained from the Hamiltonian in eq. (R1) and have been given in the previous manuscript.

$$\hat{H}_{1D}(k_x) = \frac{\hbar^2 k_x^2}{2m^*} \sigma_0 + (\Lambda_{\text{side}} k_x + \Delta_z) \sigma_z + \Lambda_{\text{top}} k_x \sigma_y \quad (\text{R1})$$

It is important to realize that the spin components σ_y and σ_z of the electron carriers are locked to the momentum k_x in opposite directions between E_+ and E_- . Thus, the + and - subscripts also correspond to the different ‘‘chirality’’ of these bands. In our theoretical model based on the Boltzmann formalism, we proposed a phenomenological parameter, α ($= \tau_+/\tau_-$, where τ_+ and τ_- are the relaxation time of electron carriers in the E_+ and E_- states, respectively). As shown in the previous manuscript, if there is asymmetry between τ_+ and τ_- (that is, $\alpha \neq 1$), the OMR is expressed as:

$$\sigma_{xx} \simeq \frac{e^2}{h} \tau_- |\lambda_{\text{side}}| \sqrt{1 + \frac{2E_F}{m^* \lambda_{\text{side}}^2}} \left[(1 - \alpha) \frac{|\lambda_{\text{side}}|}{\lambda_{\text{side}}} \frac{g\mu_B}{2E_F + m^* \lambda_{\text{side}}^2} B_z \right] \quad (\text{R2})$$

This equation can quantitatively reproduce the linear OMR results observed in our experiment when $\alpha \neq 1$, as shown in Fig. 4c and d in the main manuscript. One possible origin of the asymmetric relaxation time between E_+ and E_- can be deduced if we consider the scattering from the 1D edge channel to the 2D channel in the InAs layer, as shown in Fig. R1b,c. In the 2D channel, the spin component σ_y of the electron carriers is also locked to k_x due to the Rashba effect induced by the electric field in the z direction. Here we consider that only the scattering within the same chirality is allowed (Fig. R1c). The relaxation time of each σ_y direction (τ_+ , τ_-) should be different between the + and - chirality due to their different density of states at the Fermi level. Because the chirality + and - are only determined by the Rashba SOI, the definition and magnitude of α do not change even when reversing the z component of \mathbf{B} ($= B_z$). This leads to the appearance of OMR even when \mathbf{B}/\mathbf{M} (Fig. R2a). We note that the σ_z component of electron carriers may not be conserved in the scattering between the 1D and 2D channels because spin-flip scattering events may occur with the existence of localized spins in (Ga,Fe)Sb, which are aligned in the z direction. The current-independence of the OMR in our system can also be explained using the same framework: When reversing the current direction, σ_y in both 1D and 2D channels are flipped, and the definition of α does not change (Fig. R2b). Thus, the TRS is broken when focusing only on the 1D channel, while the non-reciprocity does not occur when considering both channels.

The magnetic proximity effect (MPE) also plays an important role in the scattering process. As we mention in our response to Comment 2-2 later, the MPE mainly affects the 2D channel by opening a gap ($= \Delta_{2D}$) between different chirality bands as shown in Fig. R1c and R2c; as the MPE is increased, Δ_{2D} is increased. As a result, the energy dispersion is altered by the MPE, which will lead to larger imbalance between τ_+ and τ_- and larger OMR (Fig. R2c).

Fig. R1 **a**, (= Fig. 4a in the main manuscript) Energy band dispersions (E_+ : purple, E_- : green) of the 1D edge channel of InAs under Rashba SOI and a magnetic field \mathbf{B} applied parallel to the z direction (the right panel corresponds to $B = 14$ T). Here, we set $m^*/m_0 = 0.08^{\text{R11}}$, $\Delta_z/B_z = 0.52$ meV/T for $g = 18^{\text{R12}}$, $m^*\lambda_{\text{side}}^2 = 0.45$ meV $^{\text{R13}}$, and $m^*\lambda_{\text{top}}^2 = 0.027$ meV $^{\text{R13}}$. **b**, (= Supplementary Figure 5a) Schematic illustration of the InAs/(Ga,Fe)Sb bilayer heterostructure near the Hall bar edge. In the (Ga,Fe)Sb layer, the magnetic moment M_z' (red arrows) near the side edge may be canted and does not effectively induce the MPE in the InAs edge channel. On the other hand, in the 2D channel (center) side, the magnetic moment M_z is aligned in the z direction due to the magnetic anisotropy of (Ga,Fe)Sb, leading to strong MPE. **c**, (= Supplementary Figure 15a) Schematic energy band dispersions (upper panel) and their Fermi surfaces (lower panel) in the k_x direction of the 1D edge (right) and 2D (left) channels in the InAs layer, where a magnetic field \mathbf{B} is applied in the z direction perpendicular to the plane. The horizontal purple dashed line indicates the Fermi level E_F . Here we consider that the scattering between the 1D and 2D channels is only allowed within the same chirality (σ_y), which are indicated by the blue and red arrows. These scattering processes have different relaxation times of τ_+ and τ_- , respectively, because of the different density of states between E_+ and E_- at E_F .

Fig. R2 a, (= Supplementary Figure 15b) Energy band dispersions when the magnetic field B is reversed. The chirality of E_+ and E_- , which is determined by the Rashba SOI, is unchanged. Therefore, $\alpha (= \tau_+/\tau_-)$ remains unchanged. **b**, (= Supplementary Figure 15c) When we flip the current I , the scattering occurs in the $-k_x$ region. In this case, $\alpha (= \tau_+/\tau_-)$ also remains unchanged, thus the OMR in the InAs/(Ga,Fe)Sb bilayer does not depend on the current direction. **c**, (= Supplementary Figure 15d) The MPE opens a gap ($= \Delta_{2D}$) in the 2D channel. This affects the DOS of each chirality in the 2D channel, which enhances the imbalance of τ_+ and τ_- and leads to larger OMR.

➤ **Corresponding revised parts in the manuscript:**

- We cited C. Xiao et al. Phys. Rev. B **101**, 201410R (2020) as ref. 10 in the revised main manuscript and Supplementary Table 1, for a comparison with our experimental results.
- To clarify the role of the spin momentum locking in the InAs channels in inducing the OMR, we added the discussion on a possible mechanism of asymmetric relaxation time of electron carriers in the E_+ and E_- states as Supplementary Note 8 in the revised Supplementary Information.
 - ✧ we added the discussion on various possible origins proposed in previous works in the main manuscript at lines 15-18 page 2: *“To explain these OMR phenomena, various possible origins were proposed, including non-trivial Berry curvature, magnetic moments and side jump mechanism¹⁰, and coexistence of spin orbit interaction (SOI) and ferromagnetic coupling in a helical magnet.¹¹”*
 - ✧ we added the discussion on a possible mechanism of asymmetric relaxation time of electron carriers in the E_+ and E_- states as Supplementary Note 8 in the revised Supplementary Information.
 - ✧ we added Fig. R1c as a new Fig. 4b.
 - ✧ we added Fig. R1c and R2a-c as a new Supplementary Fig. 15a-d.
 - ✧ we revised the following paragraphs to explain the asymmetric scattering mechanism.

At lines 9-23 page 9

(Old) *“However, at the interface with (Ga,Fe)Sb, the electron...and $D_{ds}(E_F)$ is the Fe-related density of states.”*

(New) *“However, the OMR can be induced if the relaxation times ... **Supplementary Note 8 for detailed discussions.**”*

At lines 3-4 page 10

(Old) *“Here, we set the phenomenological parameter α as $\tau_+ = \alpha\tau_-$ to express that the relaxation time is different between the two spin channels, and this difference is essentially determined by the strength of the s-d coupling V_{sd} . Under the influence of the spin-dependent scattering at the interface ($\alpha \ll 1$), the linear B_z -dependent MR appears in the term of the conductivity σ_{xx} due to the contribution of the last term in the brackets of eq. (5).”*

(New) *“Here, we set the phenomenological parameter α as $\tau_+ = \alpha\tau_-$ to express the **different relaxation time of electron carriers in the E_+ and E_- states.** Under the influence of the **strong MPE and chirality-dependent scattering** at the interface ($\alpha \ll 1$), the linear B_z -dependent MR appears in the conductivity σ_{xx} due to the contribution of the last term in the brackets of eq. (4).”*

- To illustrate the different “chirality” between the bands E_+ and E_- in the 1D channel,

- ✧ we changed the color of each energy dispersion of Fig. 4a in the main manuscript as shown in Fig. R1a.
- ✧ we added the following sentence to the caption in Fig. 4a in the main manuscript:” *In the case ... in the z direction.*”
- ✧ we added the following sentence at lines 12-16 page 8 in the main manuscript:” *It is important to note ... in the right-side graph of Fig. 4a.*”

Comment 1-4

So far the gate voltage control is not convincing with regard to removing the effect of FM substrate. As stated before, spin-orbit coupling alone could not introduce odd-parity MR, and if the author is going to claim such, they need to establish it in the absence of an FM substrate. Because of the authors’ misunderstanding of Onsager’s reciprocal relationship, their theoretical model is largely technical and lacks a clear picture of the odd-parity physics.

Our response:

As we explained in the response to comment 1-1, we do *not* claim that spatial inversion symmetry (SIS) breaking and SOI *alone* can violate the Onsager’s reciprocal theorem and yield linear odd-parity magnetoresistance (OMR). We agree with the reviewer that breaking the time reversal symmetry (TRS) is always required for observing OMR. In fact, this important role of TRS breaking in inducing the OMR is clearly demonstrated in our gating experiment shown in Fig. 3a,c in the main manuscript: When we apply positive gate voltages ($\sim 5V$) to suppress the magnetic proximity effect (MPE) in the InAs channel, the OMR decreases from 8% to less than 1%.

Furthermore, we show another piece of experimental evidence that confirms the importance of TRS breaking in yielding OMR in Fig. R3. Here we measure magnetotransport results in a nonmagnetic InAs/GaSb bilayer, *i.e.* no ferromagnetic layer is included. The OMR magnitude of the InAs/GaSb bilayer is less than 1.8% even at 14 T, which is much smaller than that in the InAs/(Ga,Fe)Sb bilayer (= 13.5% at 10 T). This experimental result clearly indicates that the TRS breaking by MPE is crucial for producing OMR.

Fig R3 (= Supplementary Figure 13) Magnetoresistance of a nonmagnetic InAs/GaSb bilayer (left side graph) and its odd component (right side graph) at 2 K. The red and blue arrows indicate the sweep direction of the magnetic field. The OMR magnitude is less than 1.8%, much smaller than that (13.5% at 10 T) in the InAs/(Ga,Fe)Sb bilayer.

➤ **Corresponding revised parts in the manuscript:**

- In order to emphasize the role of FM coupling in inducing the OMR,
 - ✧ We added Fig. R3 to the revised Supplementary Information as a new Supplementary Fig. 13.
 - ✧ We added the description at line 10-12 page 7: *“This fact is ... in Supplementary Fig. 13.”*
 - ✧ We added the description on the preparation of the InAs/GaSb reference sample in Method.

Comment 1-5

The phenomena these authors observed might have already been accounted in the literature. In a recent paper by Zyuzin PRB 104, L140407(2021), it was discussed in the presence of both magnetization and a chiral state, which has the inversion symmetry broken, the finite amount of spin-orbit coupling seems can be coupled with the ferromagnetism for the odd-parity MR. It is clear that the presence of ferromagnetism is always necessary in this mechanism of linear MR, and SOC seems serves to enhance the linear MR effect due to its influence of the band structure. The SOC enhanced odd-parity MR has been experimentally discussed in a magnetic Weyl semimetal (Jiang et al., PRL 126, 236601 (2021)). So I think this mechanism is reasonably well established both experimentally and theoretically. The authors’ experimental plan of using magnetic proximity effect actually makes the physics more ambiguous.

Our response:

We appreciate the reviewer’s suggestion of interesting models for describing the OMR. After carefully considering these models, we conclude that these cannot give an

accurate account of the OMR in our InAs/(Ga,Fe)Sb samples. According to the paper (V. Zyuzin *Phys. Rev. B* **104**, L140407 (2021)), the Hamiltonian of a 2D Rashba system with magnetic coupling is described as

$$\hat{H} = \frac{k^2}{2m} + \lambda(k_x\sigma_y - k_y\sigma_x) + M_z\sigma_z - \mu_c \quad (\text{R3})$$

where m is the effective mass, λ is the strength of the Rashba SOI in the z direction, and μ_c is the chemical potential. The OMR components are described as

$$\delta\mathbf{j} = C_1 \frac{\lambda^2\tau}{m} [(E_z M_z)\mathbf{B} + (\mathbf{E} \cdot \mathbf{B})M_z \mathbf{e}_z] + C_2 \frac{\lambda^2\tau}{8m} (M_z B_z)\mathbf{E}, \quad (\text{R4})$$

where $\delta\mathbf{j}$ is an additional electrical current corresponding to OMR, \mathbf{E} is an electric field that drives the current, \mathbf{B} is a magnetic field, \mathbf{M} is a magnetic moment, and τ is the relaxation time. C_1 and C_2 are the magnitude constants. Here, we consider the longitudinal resistance change parallel to the x direction ($=\delta j_x$) as shown in Fig. R4. In eq. (R4), when $\mathbf{B} \parallel z$, δj_x can be described as

$$\delta j_x = C_2 \frac{\lambda^2\tau}{8m} (M_z B_z) E_x \quad (\text{R5})$$

In eq. (R5), however, when $\mathbf{B} \perp \mathbf{M}$, the last term becomes an even function of \mathbf{B} . Thus, the Zyuzin's model is not appropriate to describe our experimental results. As discussed in our response to comment 1-3, the mechanism of the OMR in our InAs/(Ga,Fe)Sb bilayer heterostructures is *different from all the previously reported ones* and deserves further in-depth theoretical study.

To explain the significance of our work, we would like to explain the reason why we aim to utilize the magnetic proximity effect (MPE) for the study of OMR in the InAs/(Ga,Fe)Sb heterostructures. The most important point of using the MPE to break the TRS is that the magnetic coupling can be controlled in a wide range by electrical means such as a gate voltage. This feature allows us to systematically study the OMR while tuning the magnetic coupling strength, avoiding all the possible extrinsic factors such as difference in quality between different samples and devices. Such a high degree of controllability has never been achieved before.

Furthermore, from the viewpoint of impact and applications, the new OMR in our InAs/(Ga,Fe)Sb samples also possesses several groundbreaking advantages:

- First, our OMR is the largest ever reported, as summarized in Table R1. The large resistance change will be applicable to novel magnetic-field sensor devices.
- Second, this is the first realization of OMR in a wide range of large magnetic field (± 10 T); OMR is observed even when the magnetization \mathbf{M} is aligned with \mathbf{B} . Meanwhile, the OMR in the previous studies appeared only within a range lower than the coercive field of \mathbf{M} (that is the magnetic field range where \mathbf{M} and \mathbf{B} can be separately controlled). This fundamental difference also leads to a large dynamic range of magnetic-field sensors.
- Third, our result is the first demonstration of the gate voltage control of an OMR. As we mentioned above, our gate-voltage control experiment confirms the importance of the magnetic coupling to the OMR. In addition, the gate-controllable OMR can be widely applicable to electrical transport devices.

- Fourth, we demonstrated an unprecedentedly large and controllable OMR in the well-established III-V semiconductor heterostructure family, which is compatible with high-speed electronics and photonics devices. Unlike other OMR studies previously reported in “exotic” materials, the present OMR material system shows the excellent compatibility with the semiconductor technology is significantly important for practical device applications.

With all abovementioned virtues, we believe that the OMR in the InAs/(Ga,Fe)Sb heterostructures is unique and will have a significant impact on both experimental and theoretical studies of the OMR and related phenomena.

➤ **Corresponding revised parts in the manuscript:**

- We cited the work of V. Zyuzin PRB **104**, L140407(2021) as ref. 11 in the revised main manuscript.
- We added the following sentence to discuss the possible origins of OMR proposed in previous studies in the main manuscript at lines 15-18 page 2: *“To explain these... in a helical magnet.¹¹”*

Fig. R4 (= Fig 2d in the main manuscript) Angle dependences of the OMR magnitude $\Delta R/R_0$, where $\Delta R = (R_{23}(4\text{ T}) - R_{23}(-4\text{ T}))/2$ and $R_0 = R_{23}(0\text{ T})$. The red, green, and blue dots indicate the OMR magnitude in the xy , yz and zx rotations, respectively.

Reviewer #2:

The manuscript, titled by “Giant gate-controlled odd parity magnetoresistance in one-dimensional channels with a magnetic proximity effect”, reports an interesting phenomenon in InAs/GaFeSb heterostructure. The authors observed a giant odd parity magnetoresistance (OMR) as large as 27% in an InAs quantum well, which may be induced by a proximity effect from underlying ferromagnetic semiconductor GaFeSb. They also show that the OMR can be controlled by electrical gating either with a top-gate or separated edge gates. In addition, a theoretical analysis is proposed to explain the observed OMR.

The experimental results are very interesting and it would bring a lot attraction in the field of spintronics. However, both experiments and theoretical analysis have lots of issues to be resolved.

Our response:

We thank the reviewer for his/her encouraging and valuable comments and the high evaluation on the potential significance of our work. For answering the questions raised by the reviewer, we performed additional magnetotransport measurements, and added new data and discussions in the revised manuscript.

Comment 2-1

Regarding experiments, it seems that the OMR can be any values in the devices the authors measured, from 27% at ± 10 T, to 0.6% at ± 1 T. In particular, Fig. 2, the data in 2a and 2d are not consistent with Data in 2b. Could the authors explain what makes the OMR value varies so much in one heterostructure?

Our response:

The OMR magnitude is sensitive to some parameters related to the quality of the samples. In particular, although the devices used in Fig. 2a,d (named sample A) and Fig. 2b (sample B) have the same structure and were grown under the same conditions, the sample parameters are different: As we mentioned in the Method section, the two samples A and B have sheet electron concentrations of $2.0 \times 10^{12} \text{ cm}^{-2}$ and $1.8 \times 10^{12} \text{ cm}^{-2}$, and electron mobilities $9.4 \times 10^2 \text{ cm}^2/\text{Vs}$ and $1.9 \times 10^3 \text{ cm}^2/\text{Vs}$, respectively. We note that the sheet electron concentrations were estimated by Hall measurements, which mainly reflect the carrier concentrations of the 2D InAs channel. Although the electron concentration is quite similar, the electron mobility in sample B is twice that of sample A. This mobility difference suggests that the static electric fields, which determine the confinement potential in the 1D and 2D channels of InAs, in the two samples are different. The confinement potential sensitively affects the strength of both the PMR and the Rashba SOI in the InAs channel, consequently leading to different OMR values in these sample A and B. We note that the different confinement potentials may originate from different surface pinning effects at the top and side surfaces of the devices, which depend on the detailed conditions during the device fabrication process.

➤ Corresponding revised parts in the manuscript:

- We added descriptions to clearly indicate whether each measurement was performed using sample A or B in all the related figure captions.
- We added the following text at lines 18-19 page 3 in the main manuscript: “*We utilize ... (See Method in detail)*” to clearly state that we used two samples.
- We added the reason why the OMR magnitude varies in sample A and B to Methods section “Sample preparation and characterization”, at line 24 page 11- line 7 page 12.

Comment 2-2

On the other hand, if the authors think both 1D edge and 2D transport channels coexist, what is the current distribution ratio between them? If the edge channel is dominated in the transport, how is the OMR depending on the width of the Hall bar? Does the top gate control such ratio?

Our response:

To estimate each contribution of the 1D edge and 2D layer transport channels, we performed transport measurements on two Hall bars, D_L and D_S, with different sizes (l_{14} , l_{23} , w) = (600 μm, 200 μm, 100 μm) and (180 μm, 60 μm, 30 μm), respectively. Here w is the width of the Hall bar and l_{14} , l_{23} are the distances between terminals “1” to “4” and “2” to “3”, respectively, as shown in Fig. R5a. It is highly challenging to control the 1D and 2D conduction independently using the top gate voltage because the width of the 1D channel is too narrow. As described below, we compare the magnetotransport results and OMRs in the two Hall bars without applying a gate voltage.

As shown in Fig. R5b, temperature (T) dependence of the four-terminal resistance R_{23} shows significant difference between D_L and D_S. Because the ratios $w:l_{23}$ of D_L and D_S are the same (=1:2), R_{23} should be equal in D_L and D_S if the electrical conduction is uniform. However, the experimental $R_{23} - T$ curves differ between the two devices, which suggests that the electrical transport is non-uniform due to the coexistence of the 1D and 2D channels. As shown in the inset of Fig. R5c, the transport results can be understood by a simple resistor network model, where the resistors corresponding to the 2D (R_{2D}) and 1D (R_{1D}) channels are connected in parallel. Assuming that the 1D channel has the same width in D_L and D_S, the total resistance of the network can be expressed as:

$$(R_{23}(0\text{ T}))^{-1} = \left(r_{2D} \frac{l_{23}}{w}\right)^{-1} + (r_{1D} l_{23})^{-1} \quad (\text{R6})$$

where $R_{2D} = r_{2D}(l_{23}/w)$, and $R_{1D} = r_{1D}l_{23}$. By solving simultaneous equations for r_{2D} and r_{1D} with D_L and D_S at each temperature, we obtain separate $R_{2D} - T$ and $R_{1D} - T$ curves as shown in Fig. R5c. Note that Fig. R5c shows the case of D_L. At 3.8 K, the ratio R_{1D}/R_{2D} is 3.2, which corresponds to the current distribution ratio between the 1D and 2D channels. Thus, *2D and 1D transport channels coexist in the InAs/(Ga,Fe)Sb heterostructures, and the 1D transport does not dominate.*

One interesting observation is that the behavior of the $R_{2D} - T$ and $R_{1D} - T$ curves below 10 K agree with our model presented in Supplementary Note 2. In this Note, we argued that there is scattering between in the 1D and 2D channel, where the 2D channel has much

stronger MPE than the 1D channel. As shown in Fig. R5c, the 2D channel resistance exhibits an increase as temperature decreases below 10 K, which follows the logarithmic trend that is characteristic of the Kondo effect. This suggests strong scattering with magnetic impurities at the InAs/(Ga,Fe)Sb interface in the 2D channel. In contrast, the 1D channel resistance decreases as temperature decreases, exhibiting metallic conduction. This fact implies that the electrons in the 2D channel feel stronger MPE from the localized spins in the (Ga,Fe)Sb layer underneath, just as we expected.

The magnetotransport data of R_{23} (four-terminal resistance) are shown in Fig. R5d. The OMR magnitudes at 1 T ($= [R_{23}(1 \text{ T}) - R_{23}(-1 \text{ T})] / 2R_{23}(0 \text{ T})$) of D_L and D_s are 0.037% and 0.095%, respectively. The OMR decreases with increasing the Hall bar width w , which is reasonable considering our scenario of parallel conduction between the 1D and 2D channels: When w increases, the conduction of the 2D channel, which does not show OMR, becomes more dominant. Thus, the ratio between the resistance change due to OMR in the 1D edge channel versus the total resistance becomes smaller, leading to a smaller OMR.

This can be proved analytically as described below. Using eq. (R6), the OMR magnitude is expressed as

$$\begin{aligned} & \frac{R_{23}(1 \text{ T}) - R_{23}(-1 \text{ T})}{2R_{23}(0 \text{ T})} \\ &= \frac{\left[(R_{2D}(1 \text{ T}))^{-1} + (R_{1D}(1 \text{ T}))^{-1} \right]^{-1} - \left[(R_{2D}(-1 \text{ T}))^{-1} + (R_{1D}(-1 \text{ T}))^{-1} \right]^{-1}}{2 \left[(R_{2D}(0 \text{ T}))^{-1} + (R_{1D}(0 \text{ T}))^{-1} \right]^{-1}} \\ &= \frac{\left[(wr_{2D}(1 \text{ T}))^{-1} + (r_{1D}(1 \text{ T}))^{-1} \right]^{-1} - \left[(wr_{2D}(-1 \text{ T}))^{-1} + (r_{1D}(-1 \text{ T}))^{-1} \right]^{-1}}{2 \left[(wr_{2D}(0 \text{ T}))^{-1} + (r_{1D}(0 \text{ T}))^{-1} \right]^{-1}} \quad (\text{R7}) \end{aligned}$$

The differential of eq. (R7) with w is

$$\begin{aligned} & \frac{\partial}{\partial w} \left(\frac{R_{23}(1 \text{ T}) - R_{23}(-1 \text{ T})}{2R_{23}(0 \text{ T})} \right) \\ &= \frac{\partial}{\partial w} \left(\frac{\left[\left(h' w \right)^{-1} + g_+^{-1} \right]^{-1} - \left[\left(h' w \right)^{-1} + g_-^{-1} \right]^{-1}}{2 \left[\left(h_0 w \right)^{-1} + g_0^{-1} \right]^{-1}} \right) \\ &= - \frac{(g_- - g_+) (h_0 (h'^2 w^2 - g_+ g_-) + h' g_0 (2h' w + g_+ + g_-))}{2 (h' w + g_+)^2 (h' w + g_-)^2} \quad (\text{R8}) \end{aligned}$$

where $1/r_{1D}(\pm 1 \text{ T}) = g_{\pm}$, $1/r_{1D}(0 \text{ T}) = g_0$, $1/r_{2D}(\pm 1 \text{ T}) = h'$, and $1/r_{2D}(0 \text{ T}) = h_0$.

Note that since the 2D channel does not show the OMR component, $R_{2D}(1 \text{ T}) = R_{2D}(-1 \text{ T})$,

i.e. $r_{2D}(1 \text{ T}) = r_{2D}(-1 \text{ T}) = 1/h'$.

Here we prove that eq. (R8) is always negative at $w > 0$. For the first bracket on the numerator, since the OMR magnitude in eq. (R7) is defined as positive, $R_{1D}(1 \text{ T}) > R_{1D}(-1 \text{ T})$, *i.e.* $r_{1D}(1 \text{ T}) > r_{1D}(-1 \text{ T})$. Therefore, $g_- - g_+ > 0$. Also, for the second

bracket on the numerator, since all the parameters are positive, it is sufficient to prove $\hbar^2 w^2 - g_- g_+ > 0$. While $R_{1D}(0\text{ T})/R_{2D}(0\text{ T}) = 3.2$, the MR magnitude at $\pm 1\text{ T}$ is less than 6% as shown in Fig. R5d, which implies $R_{2D}(\pm 1\text{ T}) < R_{1D}(\pm 1\text{ T})$. Thus,

$$R_{2D}(1\text{ T})R_{2D}(-1\text{ T}) < R_{1D}(1\text{ T})R_{1D}(-1\text{ T})$$

$$\frac{(r_{2D}(1\text{ T}))^2}{w^2} < r_{1D}(1\text{ T})r_{1D}(-1\text{ T})$$

$$\hbar^2 w^2 - g_- g_+ > 0 \quad (\text{R9})$$

From the argument described above, the OMR magnitude decreases with increasing w . Thus, our model can explain the device size dependence of OMR, indicating that the 1D channel is the main origin of the OMR.

➤ **Corresponding revised parts in the manuscript:**

For clarifying the difference of the 1D and 2D transport,

- ✧ We added the discussion mentioned here and Fig. R5 as a new Supplementary Note 6 (“Supplementary Note 6 Comparison of the 1D and 2D transport via the device size effect”) and added Fig. R5 as a new Supplementary Figure 11 in the revised Supplementary Information.
- ✧ We added the following text at lines 7-9 page 6 in the main manuscript: “*This fact is ... in Supplementary Note 6.*”

Fig. R5 (= Supplementary Figure 11) **a**, Optical microscope image (same as Fig. 1b in the main manuscript) of the Hall bar device DL. Here, w and l_{23} indicate the width and the distance between “2” and “3” electrodes, respectively. **b**, Temperature (T) dependence of $R_{23}(0\text{ T})$ of device DL (blue) and device Ds (green). **c**, T dependence of R_{2D} (cyan) and R_{1D} (purple) in DL. The inset shows the schematic resistor network representing R_{23} . The 2D resistance R_{2D} has the width w and length l_{23} , and the 1D resistance R_{1D} has the same length. The black dashed line at $T < 10\text{ K}$ is the fitting result using a logarithmic function, which is characteristic of the Kondo effect ($R_{2D} = R_{c0} - R_{c1} \ln T$, where R_{c0} and R_{c1} are fitting parameters). **d**, Normalized magnetoresistance by the zero-field resistance, $[R_{23}(B) - R_{23}(0\text{ T})]/R_{23}(0\text{ T})$, of DL (blue) and Ds (green).

Comment 2-3

What is the Hall bar orientation respect to the film crystallographic direction? Does such OMR depend on the Hall bar orientation?

Our response:

The Hall bar orientation (current direction) presented in this paper is always along the $[\bar{1}10]$ axis of the GaAs substrate. In order to investigate the effect of crystal orientation, we fabricate a Hall bar device aligned along the $[110]$ direction and compare its magnetotransport data with those of the Hall bar device aligned along the $[\bar{1}10]$ direction (The device aligned along $[\bar{1}10]$ is the same as D_L in the response to comment 2-2). Figure R6a shows temperature dependence of the four-terminal resistance R_{23} in the two devices where the current I is applied parallel to $[110]$ and $[\bar{1}10]$, respectively. The resistance at each T differs between two Hall bars with different orientations. Also, the magnetoresistance measurements exhibit different OMR magnitude as shown in Fig. R6b. The OMR magnitude ($= [R_{23}(1\text{ T}) - R_{23}(-1\text{ T})] / 2R_{23}(0\text{ T})$) of $[110]$ and $[\bar{1}10]$ are 0.065% and 0.036%, respectively.

According to the previous study of the Rashba and Dresselhaus effects of InAs/GaSb heterostructures (Knox et al., *Phys. Rev. B* 98, 155323 (2018))^{R6}, the Dresselhaus effect is relatively small, less than one-sixth of the Rashba effect. Thus, it is unlikely that the differences in R_{23} and the OMR magnitude originate from the Dresselhaus effect. One possible reason is the anisotropic distribution of the Fe atoms in the (Ga,Fe)Sb layer with an Fe content of $\sim 20\%$ along the two directions, $[110]$ and $[\bar{1}10]$. In heavily Fe-doped (Ga,Fe)Sb (Fe $> \sim 20\%$) such as that in our InAs/(Ga,Fe)Sb samples, it is known that spinodal decomposition occurs, leading to fluctuation in the local Fe concentration in the host GaSb crystal (Goel et al., *J. Appl. Phys.* 127, 023904 (2020))^{R7}. This Fe-rich (Ga,Fe)Sb regions can favorably form in one direction, $[110]$ or $[\bar{1}10]$ (Yuan et al., *Phys. Rev. Mater.* 2, 114601 (2018))^{R8}. If this is the case, it can lead to different strength of the magnetic proximity effect (MPE) when the electron carriers in InAs flow in different directions, which will lead to anisotropic OMR.

Fig. R6 (Supplementary Figure 6) **a**, Temperature (T) dependence of $R_{23}(0\text{T})$ with $I // [110]$ (red) and $I // [\bar{1}10]$ (blue). **b**, (upper) Normalized magnetoresistance by the zero-field resistance, $[R_{23}(B) - R_{23}(0\text{ T})] / R_{23}(0\text{ T})$, with $I // [110]$ (red) and $I // [\bar{1}10]$ (blue).

➤ **Corresponding revised parts in the manuscript:**

For showing the difference of the transport data between the two Hall bar orientations,

- ✧ We added the discussion mentioned here and Fig. R6 as a new Supplementary Note 3 (“Supplementary Note 3. Hall-bar-orientation dependence of OMR”) and added Fig. R6 as a new Supplementary Figure 6 in the revised Supplementary Information.
- ✧ We added the following text at lines 2-4 page 5 in the main manuscript: “*Also, we find ... (See Supplementary Note 3).*”

Comment 2-4

The oscillation frequency of SdH represents what channel of conductivity?

Our response:

The Shubnikov de Haas (SdH) oscillations with various magnetic field (**B**) directions represent the 2D transport. Figure R7 shows the magnetoresistance of each rotation angle β in the yz plane. If the SdH comes from the 1D channel, this β rotation does not affect the formation of Landau levels due to the equal contribution of the magnetic field to the 1D system. However, the SdH oscillation disappears when the magnetic field is parallel to the y direction. Thus, the SdH oscillation is originated from the 2D transport in the InAs layer.

➤ **Corresponding revised parts in the manuscript:**

To show the Hall bar orientation dependence of OMR,

- ✧ We added the following text at lines 13-14 page 4 in the main manuscript: “*Also for (iii)... (See Supplementary Fig. 1).*”
- ✧ We added Fig. R7 as a new Supplementary Fig. 1.

Fig. R7 (= Supplementary Fig. 1) Magnetoresistance of sample A in each β angle from 0° to 90° . The inset shows the definition of β in the yz plane.

Comment 2-5

What is the electron density, Fermi energy and mobility obtained from SdH oscillation?

Our response:

In sample A, from the SdH oscillation with a frequency of 41 T (Fig. R8a), the electron carrier density is estimated to be $2.02 \times 10^{12} \text{ cm}^{-2}$, which agrees well with $1.96 \times 10^{12} \text{ cm}^{-2}$ estimated from the Hall measurement. At this electron density, our calculation shows that the Fermi level is located at 67.8 meV above the conduction band bottom of InAs. Here, the electron effective mass of InAs is set to be $0.07m_0$ (m_0 is the electron mass in a vacuum), which is obtained by our latest SdH study. Also, we performed the Dingle plot^{R9} to estimate the quantum mobility as shown in Fig. R8b. The quantum mobility is $2070 \text{ cm}^2/\text{Vs}$.

Fig R8 (= Supplementary Fig. 14) **a**, Oscillating component ΔR_{osc} of sample A extracted from the magnetotransport data shown in Fig. 1c in the main manuscript. The background signal is subtracted from the raw data by using a third polynomial function. **b**, Dingle plot from the data in **a**. A_{osc} is the peak value of the SdH oscillation. R_T is the temperature reduction factor: $R_T = \sinh X / X$, $X = 2\pi k_B T / \hbar \omega_c$. Here, k_B is Boltzmann's constant, and $\omega_c (= eB/m^*)$ is the cyclotron angular frequency. The black line indicates the fitting of the Dingle plot.

➤ Corresponding revised parts in the manuscript:

For more information about the SdH oscillation data,

- ✧ We added the information of the quantum mobility of sample A to section “Sample preparation and characterization” in Method.
- ✧ We added the Fig. R8 as a new Supplementary Fig. 14.

Comment 2-6

The surface pinning also happens on InAs top surface, why is the edge (2 micron from KFM) is so different from top surface of InAs? Is this the nano fabrication effect?

Our response:

It is likely that the environment of the top surface is different from that of edge surface. The edge surface is surrounded by SiO_2 deposited by sputtering for passivation, while the top surface was not treated with anything. This may lead to different pinning effects in these two interfaces.

Also, we note that the potential profile at the topmost InAs surface detected by KFM might be different from the one at several-atomic-layer depth below the surface. This is because of a screening effect from a large amount of charged surface states (top and side surfaces), which are common at InAs surfaces. Thus, the potential profile change along the y direction measured by the KFM tip is much *milder* than the real confinement potential at the edge of the InAs channel^{R10}. As a result, the potential profile at the top surface shown in Fig. R9 might largely exaggerate the width of the triangular potential at the bulk InAs side surface, which should be much less than $2\ \mu\text{m}$. Therefore, we consider that the static electric field in the y and z directions should have same order of magnitude.

Fig. R9 (= Supplementary Fig. 10) **a**, Schematic configuration of the KFM measurement. **b**, KFM (blue solid line) and AFM (green dashed line) results obtained from the y direction sweep. The red shaded area is the place where the tip goes through the edge of the sample.

➤ **Corresponding revised parts in the manuscript:**

In order to clarify what KFM signal represents,

- ✧ we revised the Method section “Work function measurements by Kelvin probe force microscopy (KFM)”.
- ✧ we deleted the citation #31 and #32, which is not directly related with the abovementioned phenomenon.

At line 16 page 13- line 2 page 14

(Old) “We note that, in systems with large amount of charged surface states, the in-plane distance and energy obtained by KFM are usually over- and underestimated, respectively.^{30,31,32} Therefore, the KFM results in the KFM results in Supplementary Fig. 4b largely exaggerate the width of the triangular potential at the InAs side surface, which should be less than $1\ \mu\text{m}$.”

(New) *“We note that, the potential profile at the topmost InAs surface detected by KFM might be different from the one at several-atomic-layer depth below the surface. This is because of a screening effect from a large amount of charged surface states (top and side surfaces), which are common at InAs surfaces. Thus, the potential profile change along the y direction measured by the KFM tip is much milder than the real confinement potential at the edge of the InAs channel.³⁰ As a result, the potential profile at the top surface shown in Supplementary Fig. 10b might largely exaggerate the width of the triangular potential at the bulk InAs side surface, which should be much less than 2 μm . Therefore, we consider that the static electric field in the y and z directions should have same order of magnitude.”*

Comment 2-7

What is the real morphology profile of the edge of the InAs/GaFeSb Hall bar?

Our response:

As previously shown in Fig. R9b (= Supplementary Fig. 10 in the revised Supplementary Information), the morphology is presented in the green dashed line obtained by atomic force microscopy (AFM). The morphology profile (2D mapping) is shown in Fig. R10. The edge of the side surface is clearly observed.

Fig. R10 AFM mapping of the InAs/(Ga,Fe)Sb bilayer heterostructure (sample A) .

Regarding theoretical analysis, two key issues the authors should clarify.

Comment 2-8

1. The structure inversion symmetry is broken at the interface between InAs and GaFeSb. What effect is such breaking on the OMR analysis?

Our response:

The structure inversion symmetry at the interface yields the spin momentum locking of the y component of spin ($= \sigma_y$) due to the Rashba effect. This plays an important role for the asymmetric scattering, leading to the OMR. The detailed discussion is described below.

As shown in eq. (R10) ($=$ eq. (1) in the previous/revised main manuscript), the Hamiltonian for the 1D edge channel of InAs in our InAs/(Ga,Fe)Sb heterostructures is given by:

$$\hat{H}_{1D}(k_x) = \frac{\hbar^2 k_x^2}{2m^*} \sigma_0 + (\Lambda_{\text{side}} k_x + \Delta_z) \sigma_z + \Lambda_{\text{top}} k_x \sigma_y \quad (\text{R10})$$

where k_x is the wavenumber along the x direction, m^* is the effective mass of electrons, $\Lambda_{\text{top(side)}} (= \hbar \lambda_{\text{top(side)}})$ is the effective Rashba spin orbit interaction (SOI) due to the built-in potential at the top (side edge) surface, \hbar is Dirac's constant, $\Delta_z (= g\mu_B B_z)$ is the Zeeman splitting due to an applied magnetic field along the z -axis (B_z), σ_i ($i = x, y, z$) are the elements of the Pauli matrix that act on the electron spin degree of freedom, and σ_0 is the identity matrix. The structure inversion symmetry breaking in the z direction results in the last term of the Rashba effect, $\Lambda_{\text{top}} k_x \sigma_y$. Figure R11a illustrates the energy band dispersions of the 1D edge channel of InAs, where there are two branches E_+ and E_- . As the results of the Rashba SOI in the top (z direction) and side (y direction) surface of the InAs edge, the spin components σ_y and σ_z of electron carriers are locked to the momentum k_x in opposite directions between the E_+ and E_- states. Thus, the $+$ and $-$ subscripts mean the difference of the ‘‘chirality’’ of these bands. This different ‘‘chirality’’ of E_+ and E_- is important in generating the OMR in our InAs/(Ga,Fe)Sb heterostructures, as explained below.

In our theoretical model based on the Boltzmann formalism, we proposed a phenomenological parameter, $\alpha (= \tau_+/\tau_-)$, where τ_+ and τ_- are the relaxation time of electrons in the E_+ and E_- states, respectively). As shown in the previous manuscript, if there is asymmetry between τ_+ and τ_- (that is, $\alpha \neq 1$), the longitudinal conductivity σ_{xx} can be expressed as:

$$\sigma_{xx} \simeq \frac{e^2}{h} \tau_- |\lambda_{\text{side}}| \sqrt{1 + \frac{2E_F}{m^* \lambda_{\text{side}}^2}} \left[(1 - \alpha) \frac{|\lambda_{\text{side}}|}{\lambda_{\text{side}}} \frac{g\mu_B}{2E_F + m^* \lambda_{\text{side}}^2} B_z \right] \quad (\text{R11})$$

This equation can quantitatively reproduce the linear OMR results observed in our experiment when $\alpha \neq 1$, as shown in Fig. 4c and d in the main manuscript. One possible origin of the asymmetric relaxation time between E_+ and E_- can be deduced if we consider the scattering from the 1D edge channel to the 2D channel in the InAs layer, as shown in Fig. R11b,c. In the 2D channel, the spin component σ_y of the electron carriers is also locked to k_x due to the Rashba effect induced by the electric field in the z direction. Here we consider that only the scattering within the same chirality is allowed (Fig. R11c). The relaxation time of each σ_y direction (τ_+, τ_-) should be different between the $+$ and $-$ chirality due to their different density of states at the Fermi level. Because the chirality $+$ and $-$ are only determined by the Rashba SOI, the definition and magnitude of α do not change even when reversing the z component of \mathbf{B} ($= B_z$). This leads to the appearance of OMR even when $\mathbf{B} // \mathbf{M}$ (Fig. R12a). We note that the σ_z component of electron carriers may not be conserved in the scattering between the 1D and 2D channels because spin-flip scattering events may occur with the existence of localized spins in (Ga,Fe)Sb, which are aligned in the z direction. The current-

independence of the OMR in our system can also be explained using the same framework: When reversing the current direction, σ_y in both 1D and 2D channels are flipped, and the definition of α does not change (Fig. R12b). Thus, the TRS is broken when focusing only on the 1D channel, while the non-reciprocity does not occur when considering both channels.

Fig. R11 **a**, (= Fig. 4a in the main manuscript) Energy band dispersions (E_+ : purple, E_- : green) of the 1D edge channel of InAs under Rashba SOI and a magnetic field \mathbf{B} applied parallel to the z direction (the right panel corresponds to $B = 14$ T). Here, we set $m^*/m_0 = 0.08^{\text{R11}}$, $\Delta_z/B_z = 0.52$ meV/T for $g = 18^{\text{R12}}$, $m^*\lambda_{\text{side}}^2 = 0.45$ meV $^{\text{R13}}$, and $m^*\lambda_{\text{top}}^2 = 0.027$ meV $^{\text{R13}}$. **b**, (= Supplementary Figure 5a) Schematic illustration of the InAs/(Ga,Fe)Sb bilayer heterostructure near the Hall bar edge. In the (Ga,Fe)Sb layer, the magnetic moment M_z' (red arrows) near the side edge may be canted and does not effectively induce the MPE in the InAs edge channel. On the other hand, in the 2D channel (center) side, the magnetic moment M_z is aligned in the z direction due to the magnetic anisotropy of (Ga,Fe)Sb, leading to strong MPE. **c**, (= Supplementary Figure 15a) Schematic energy band dispersions (upper panel) and their Fermi surfaces (lower panel) in the k_x direction of the 1D edge (right) and 2D (left) channels in the InAs layer, where a magnetic field \mathbf{B} is applied in the z direction perpendicular to the plane. The horizontal purple dashed line indicates the Fermi level E_F . Here we consider that the scattering between the 1D and 2D channels is only allowed within the same chirality (σ_y), which are indicated by the blue and red arrows. These scattering processes have different relaxation times of τ_+ and τ_- , respectively, because of the different density of states between E_+ and E_- at E_F .

The magnetic proximity effect (MPE) also plays an important role in the scattering process. As we mention in our response to Comment 2-2 later, the MPE mainly affects the 2D channel by opening a gap ($=\Delta_{2D}$) between different chirality bands as shown in Fig. R11c and R12c; as the MPE is increased, Δ_{2D} is increased. As a result, the energy dispersion is altered by the MPE, which will lead to larger imbalance between τ_+ and τ_- and larger OMR (Fig. R12c).

Fig. R12 a, (= Supplementary Figure 15b) Energy band dispersions when the magnetic field B is reversed. The chirality of E_+ and E_- , which is determined by the Rashba SOI, is unchanged. Therefore, $\alpha (= \tau_+/\tau_-)$ remains unchanged. **b**, (= Supplementary Figure 15c) When we flip the current I , the scattering occurs in the $-k_x$ region. In this case, $\alpha (= \tau_+/\tau_-)$ also remains unchanged, thus the OMR in the InAs/(Ga,Fe)Sb bilayer does not depend on the current direction. **c**, (= Supplementary Figure 15d) The MPE opens a gap ($=\Delta_{2D}$) in the 2D channel. This affects the DOS of each chirality in the 2D channel, which enhances the imbalance of τ_+ and τ_- and leads to larger OMR.

➤ **Corresponding revised parts in the manuscript:**

- We cited C. Xiao et al. PRB **101**, 201410R (2020) as ref. 10 in the revised main manuscript and Supplementary Table 1, for a comparison with our experimental results.
- To clarify the role of the spin momentum locking in the InAs channels in inducing the OMR, we added the discussion on a possible mechanism of asymmetric relaxation time of carriers in the E_+ and E_- states as Supplementary Note 8 in the revised Supplementary Information.
 - ✧ we added the discussion on various possible origins proposed in previous works in the main manuscript at lines 15-18 page 2: *“To explain these OMR phenomena, various possible origins were proposed, including non-trivial Berry curvature, magnetic moments and side jump mechanism¹⁰, and coexistence of spin orbit interaction (SOI) and ferromagnetic coupling in a helical magnet.¹¹”*
 - ✧ we added the discussion on a possible mechanism of asymmetric relaxation time of electron carriers in the E_+ and E_- states as Supplementary Note 8 in the revised Supplementary Information.
 - ✧ we added Fig. R11c as a new Fig. 4b.
 - ✧ we added Fig. R11c, R12a-c as a new Supplementary Fig. 15a-d.
 - ✧ we revised the following paragraphs to explain the asymmetric scattering mechanism.

At lines 9-23 page 9

- (Old) *“However, at the interface with (Ga,Fe)Sb, the electron...and $D_{ds}(E_F)$ is the Fe-related density of states.”*
- (New) *“However, the OMR can be induced if the relaxation times ... Supplementary Note 8 for detailed discussions).”*

At lines 3-4 page 10

- (Old) *“Here, we set the phenomenological parameter α as $\tau_+ = \alpha\tau_-$ to express that the relaxation time is different between the two spin channels, and this difference is essentially determined by the strength of the s-d coupling V_{sd} . Under the influence of the spin-dependent scattering at the interface ($\alpha \ll 1$), the linear B_z -dependent MR appears in the term of the conductivity σ_{xx} due to the contribution of the last term in the brackets of eq. (5).”*
- (New) *“Here, we set the phenomenological parameter α as $\tau_+ = \alpha\tau_-$ to express the different relaxation time of electron carriers in the E_+ and E_- states. Under the influence of the strong MPE and chirality-dependent scattering at the interface ($\alpha \ll 1$), the linear B_z -dependent MR appears in the conductivity σ_{xx} due to the contribution of the last term in the brackets of eq. (4).”*

- To illustrate the different “chirality” between the bands E_+ and E_- in the 1D channel,
 - ✧ we changed the color of each energy dispersion of Fig. 4a in the main manuscript as shown in Fig. R11a.

- ✧ we added the following sentence to the caption in Fig. 4a in the main manuscript:” *In the case ... in the z direction.*”
- ✧ we added the following sentence at lines 12-16 page 8 in the main manuscript:” *It is important to note ... in the right-side graph of Fig. 4a.*”
- We added the suffix of “x” to k (wavevector) in eq. (1) in order to clarify the direction of the 1D channel like eq. (R10).

[Q2-9]

2. In Fig. 4a, why do both the side Rashba effect and magnetic field (Zeeman effect) make electron spin degenerate in z direction?

Our response:

The magnetic field *lifts spin degeneracy*, which is known as the Zeeman effect. The Zeeman effect affects the spin component which is parallel to the magnetic field. The Rashba effect at the side of the InAs edge (A_{side}) comes from the electric field E_{sur} that is formed due to the triangular potential and points to the y direction. According to the Lorentz transformation, an electric field works as an effective magnetic field \mathbf{B}' for moving electrons: $\mathbf{B}' = -\mathbf{v} \times \mathbf{E}/c^2$, where \mathbf{v} is the velocity, \mathbf{E} is static electric field, and c is the light velocity. Thus, the electric field in the y direction generates an effective magnetic field (Rashba field) along the z direction when the electron carriers flow in x direction.

This Rashba magnetic field has the same effect as that of the external magnetic field applied parallel to the z direction. Therefore, both the side Rashba effect and the external magnetic field resolve the electron spin degeneracy in the z direction.

Reviewer #3:

Electron and spin transport in InAs/GaSb (a quantum spin Hall system) has attracted large attention in recent years. In this work, a more interesting system InAs/GaFeSb (ferromagnetic) was investigated. In this system, a very interesting and rare OMR was observed. The authors performed well-designed experiments. The model is also reasonable. I recommend to publish this paper in Nature Communications after the authors tackle several minor points.

Our response:

We thank the reviewer for his/her encouraging comment, high evaluation on our work, and recommending publication in *Nature Communications*. In the following, we would like to answer the questions point by point.

Comment 3-1

The Hall measurement results of the system should be very interesting. I believe that every reader of this paper wants to see the Hall measurement results, which can provide a lot of important information to understand this interesting OMR effect and improve the quality of the paper. Could the authors show the results somewhere in supplementary?

Our response:

We show the Hall resistance data of device D1 ($= V_{26}/I_{14}$) in Supplementary Fig. 9 (same data are shown in Fig. R14 and R13 below in this response letter). From Fig. R14a, one can see that the Hall resistances are almost independent of the current I in the whole range of from 100 nA to 100 μ A. In contrast, the current dependence of OMR in the same sample has a step-like change at 200 nA, as shown in Fig. R14b. This fact indicates that the origin of the Hall effect is completely different from that of OMR. In addition, we also investigate the Hall resistance in device D2 with separate gate electrodes (Fig. R13a) when we apply a separated gate voltage V_{g1} . Figure R13b shows the Hall resistance at various V_{g1} , where V_{g1} can change the contribution of the 1D edge channel transport in InAs. Again, as shown in Fig. R13b and c, the OMR and the Hall resistance with applying the gate voltage are different: The OMR in two-terminal measurement (odd component R_{14}^{odd} of $R_{14} = V_{14}/I_{14}$) exhibits the polarity change with V_{g1} (lower panel of Fig. R13c), while the Hall resistance (V_{26}/I_{14}) shows a negative slope in all V_{g1} (Fig. R13b). Thus, we conclude that the origins of the OMR and Hall resistance are different; our OMR is originated from the 1D edge channel transport in InAs and asymmetric scattering ($\alpha = \tau_+/\tau_- \neq 1$, see eq.(4) in the revised main manuscript and eq.(R2) and (R11) in this response letter). The detailed Hall resistance data and related discussion are added as Supplementary Note 5 and Supplementary Fig. 9 in the revised Supplementary Information.

Figure R14 **a**, (= Supplementary Fig. 9a) Hall resistance (V_{26}/I_{14}) of device D1 at various current measured by the lock-in technique with 5261 Hz at 3.5 K. **b**, (= Fig. 2a in the main manuscript) Current (I) dependence of the OMR magnitude $\Delta R/R_0$ ($= [(R_{23}(1\text{ T}) - R_{23}(-1\text{ T}))/2]/R_0$) measured in R_{23} at 3.5 K. R_0 is the zero field resistance. The R_0 and $\Delta R/R_0$ jump up at I_c ($= 200$ nA) simultaneously (see Fig. 2a in the main manuscript).

Figure R13 **a**, (= Fig. 3b in the main manuscript) Optical microscopy image of the Hall-bar field-effect transistor (FET) device D2 with separated gate electrodes G_1 and G_2 , respectively. The dashed line indicates the outline of the Hall bars. The light and dark yellow parts are the Au pads of gate and Hall-bar electrodes, respectively. **b**, (= Supplementary Fig. 9b) Hall resistance (V_{26}/I_{14}) vs. perpendicular magnetic field B of D2 measured at various gate voltage V_{g1} (applied to gate G_1) at 2 K with 1μ A. **c**, (= Fig. 3d in the main manuscript) MR results of device D2 (top panel) and the odd components (bottom panel) at 2 K. R_{14} is the resistance measured between terminals 1 and 4 (two-terminal measurement). The MR results at $V_{g1} = -7, 0, +7$ V are shown in green, black and blue lines, respectively.

Comment 3-2

The band structure shown in Figure 4a induces spin momentum locking in the edge transport. Could spin momentum locking has some relationship with this interesting OMR?

Our response:

The spin momentum locking due to the Rashba spin orbit interaction (SOI) plays an important role in generating the OMR in our InAs/(Ga,Fe)Sb system, as described below.

As shown in eq. (R12) (= eq. (1) in the main manuscript), the Hamiltonian for the 1D edge channel of InAs in our system is given by:

$$\hat{H}_{1D}(k_x) = \frac{\hbar^2 k_x^2}{2m^*} \sigma_0 + (\Lambda_{\text{side}} k_x + \Delta_z) \sigma_z + \Lambda_{\text{top}} k_x \sigma_y \quad (\text{R12})$$

where k_x is the wavenumber along the x direction, m^* is the effective mass of electrons, $\Lambda_{\text{top(side)}} (= \hbar \lambda_{\text{top(side)}})$ is the effective Rashba SOI due to the built-in potential at the top (side) edge) surface, \hbar is Dirac's constant, $\Delta_z (= g\mu_B B_z)$ is the Zeeman splitting due to an applied magnetic field along the z -axis (B_z), σ_i ($i = x, y, z$) are the elements of the Pauli matrix that act on the electron spin degree of freedom, and σ_0 is the identity matrix. The structure inversion symmetry breaking in the z direction is expressed in the last term of the Rashba effect. Figure R15a illustrates the energy band dispersions of the 1D edge channel of InAs, where there are two branches E_+ and E_- , which are the results of the Rashba spin orbit interaction (SOI) in the top (z direction) and side (y direction) surface of the InAs edge (see Fig.4a in the manuscript or Fig. R15 below).

It is important to realize that the spin components σ_y and σ_z of the electron carriers are locked to the momentum k_x in opposite directions between E_+ and E_- . Thus, the + and – subscripts also correspond to the different “chirality” of these bands. In our theoretical model based on the Boltzmann formalism, we proposed a phenomenological parameter, $\alpha (= \tau_+/\tau_-)$, where τ_+ and τ_- are the relaxation time of electron carriers in the E_+ and E_- states, respectively). As shown in the previous manuscript, if there is asymmetry between τ_+ and τ_- (that is, $\alpha \neq 1$), the OMR is expressed as:

$$\sigma_{xx} \approx \frac{e^2}{h} \tau_- |\lambda_{\text{side}}| \sqrt{1 + \frac{2E_F}{m^* \lambda_{\text{side}}^2}} \left[1 + \alpha + (1 - \alpha) \frac{|\lambda_{\text{side}}|}{\lambda_{\text{side}}} \frac{g\mu_B}{2E_F + m^* \lambda_{\text{side}}^2} B_z \right] \quad (\text{R5})$$

This equation can quantitatively reproduce the linear OMR results observed in our experiment when $\alpha \neq 1$, as shown in Fig. 4c and d in the main manuscript. One possible origin of the asymmetric relaxation time between E_+ and E_- can be deduced if we consider the scattering from the 1D edge channel to the 2D channel in the InAs layer, as shown in Fig. R15b,c. In the 2D channel, the spin component σ_y of the electron carriers is also locked to k_x due to the Rashba effect induced by the electric field in the z direction. Here we consider that only the scattering within the same chirality is allowed (Fig. R15c). The relaxation time of each σ_y direction (τ_+ , τ_-) should be different between the + and – chirality due to their different density of states at the Fermi level. Because the chirality + and – are only determined by the Rashba SOI, the definition and magnitude of α do not change even when reversing the z component of \mathbf{B} ($= B_z$). This leads to the appearance of OMR even when \mathbf{B}/\mathbf{M} (Fig. R16a). We note that the σ_z component of electron carriers may not be conserved in the scattering

between the 1D and 2D channels because spin-flip scattering events may occur with the existence of localized spins in (Ga,Fe)Sb, which are aligned in the z direction. The current-independence of the OMR in our system can also be explained using the same framework: When reversing the current direction, σ_y in both 1D and 2D channels are flipped, and the definition of α does not change (Fig. R16b). Thus, the TRS is broken when focusing only on the 1D channel, while the non-reciprocity does not occur when considering both channels.

The magnetic proximity effect (MPE) also plays an important role in the scattering process. As we mention in our response to Comment 2-2 later, the MPE mainly affects the 2D channel by opening a gap ($=\Delta_{2D}$) between different chirality bands as shown in Fig. R15c and R16c; as the MPE is increased, Δ_{2D} is increased. As a result, the energy dispersion is altered by the MPE, which will lead to larger imbalance between τ_+ and τ_- and larger OMR (Fig. R16c).

➤ **Corresponding revised parts in the manuscript:**

- To clarify the role of the spin momentum locking in the InAs channels in inducing the OMR, we added the discussion on a possible mechanism of asymmetric relaxation time of electron carriers in the E_+ and E_- states as Supplementary Note 8 in the revised Supplementary Information.
 - ✧ we added the discussion on various possible origins proposed in previous works in the main manuscript at lines 15-18 page 2: *“To explain these OMR phenomena, various possible origins were proposed, including non-trivial Berry curvature, magnetic moments and side jump mechanism¹⁰, and coexistence of spin orbit interaction (SOI) and ferromagnetic coupling in a helical magnet.¹¹”*
 - ✧ we added the discussion on a possible mechanism of asymmetric relaxation time of electron carriers in E_+ and E_- states as Supplementary Note 8 in the revised Supplementary Information.
 - ✧ we added as Fig. R15c as a new Fig. 4b.
 - ✧ we added Fig. R15c, R16a-c as a new Supplementary Fig. 15a-d.
 - ✧ we revised the following paragraphs to explain the asymmetric scattering mechanism.

At lines 9-23 page 9

- (Old) *“However, at the interface with (Ga,Fe)Sb, the electron...and $D_{ds}(E_F)$ is the Fe-related density of states.”*
- (New) *“However, the OMR can be induced if the relaxation times ... Supplementary Note 8 for detailed discussions).”*

At lines 3-4 page 10

- (Old) *“Here, we set the phenomenological parameter α as $\tau_+ = \alpha\tau_-$ to express that the relaxation time is different between the two spin channels, and this difference is essentially determined by the strength of the s - d coupling V_{sd} . Under the influence of the spin-dependent scattering at the interface ($\alpha \ll 1$), the linear B_z -dependent MR appears in the term of the conductivity σ_{xx} due to the contribution of the last term in the brackets of eq. (5).”*

(New) “Here, we set the phenomenological parameter α as $\tau_+ = \alpha\tau_-$ to express the *different relaxation time of electron carriers in the E_+ and E_- states.* Under the influence of the *strong MPE and chirality-dependent scattering at the interface ($\alpha \ll 1$), the linear B_z -dependent MR appears in the conductivity σ_{xx} due to the contribution of the last term in the brackets of eq. (4).”*

- To illustrate the different “chirality” between the bands E_+ and E_- in the 1D channel,
 - ✧ we changed the color of each energy dispersion of Fig. 4a in the main manuscript as shown in Fig. R15a.
 - ✧ we added the following sentence to the caption in Fig. 4a in the main manuscript:” *In the case ... in the z direction.*”
 - ✧ we added the following sentence at lines 12-16 page 8 in the main manuscript:” *It is important to note ... in the right-side graph of Fig. 4a.*”

Fig. R15 **a**, (= Fig. 4a in the main manuscript) Energy band dispersions (E_+ : purple, E_- : green) of the 1D edge channel of InAs under Rashba SOI and a magnetic field \mathbf{B} applied parallel to the z direction (the right panel corresponds to $B = 14$ T). Here, we set $m^*/m_0 = 0.08^{\text{R11}}$, $\Delta_z/B_z = 0.52$ meV/T for $g = 18^{\text{R12}}$, $m^*\lambda_{\text{side}}^2 = 0.45$ meV $^{\text{R13}}$, and $m^*\lambda_{\text{top}}^2 = 0.027$ meV $^{\text{R13}}$. **b**, (= Supplementary Figure 5a) Schematic illustration of the InAs/(Ga,Fe)Sb bilayer heterostructure near the Hall bar edge. In the (Ga,Fe)Sb layer, the magnetic moment M_z' (red arrows) near the side edge may be canted and does not effectively induce the MPE in the InAs edge channel. On the other hand, in the 2D channel (center) side, the magnetic moment M_z is aligned in the z direction due to the magnetic anisotropy of (Ga,Fe)Sb, leading to strong MPE. **c**, (= Supplementary Figure 15a) Schematic energy band dispersions (upper panel) and their Fermi surfaces (lower panel) in the k_x direction of the 1D edge (right) and 2D (left) channels in the InAs layer, where a magnetic field \mathbf{B} is applied in the z direction perpendicular to the plane. The horizontal purple dashed line indicates the Fermi level E_F . Here we consider that the scattering between the 1D and 2D channels is only allowed within the same chirality (σ_y), which are indicated by the blue and red arrows. These scattering processes have different relaxation times of τ_+ and τ_- , respectively, because of the different density of states between E_+ and E_- at E_F .

Fig. R16 a, (= Supplementary Figure 15b) Energy band dispersions when the magnetic field B is reversed. The chirality of E_+ and E_- , which is determined by the Rashba SOI, is unchanged. Therefore, $\alpha (= \tau_+/\tau_-)$ remains unchanged. **b**, (= Supplementary Figure 15c) When we flip the current I , the scattering occurs in the $-k_x$ region. In this case, $\alpha (= \tau_+/\tau_-)$ also remains unchanged, thus the OMR in the InAs/(Ga,Fe)Sb bilayer does not depend on the current direction. **c**, (= Supplementary Figure 15d) The MPE opens a gap ($= \Delta_{2D}$) in the 2D channel. This affects the DOS of each chirality in the 2D channel, which enhances the imbalance of τ_+ and τ_- and leads to larger OMR.

Comment 3-3

How do you define the spin-up and spin-down electrons? As the spin is also related to momentum direction, can we define the spin-up and spin-down based on the magnetization direction?

Our response:

The quantized axis for spin is set to be the z direction (σ_z), and the definition of the spin direction is determined by the (Ga,Fe)Sb's perpendicular magnetization. In the 2D channel, the inner and outer energy bands can be labeled by the σ_z spin component (left side graph of Fig. R15c). On the other hand, σ_z in the 1D edge channel of InAs is not a good quantum number due to the Rashba spin orbit interaction induced by the structure inversion symmetry breaking in the y direction. However, as we mention in Supplementary Note 2 and 6 (and our response to Comment 2-2 in this response letter), we consider that the MPE mainly occurs in the 2D channel. When considering MPE in the 2D InAs channel, it is appropriate to define the spin up and down based on the magnetization of the (Ga,Fe)Sb layer.

Also, in the previous manuscript, we drew arrows for representing the spin direction in the curves of E_+ and E_- in Fig. 4a. These arrows represented the spin components not in the perpendicular direction (σ_z) but in *the in-plane direction* (σ_y); however this representation was confusing, thus we changed the expression in the revised manuscript, as shown in Fig. 4a and Fig. R15c as described in our response to comment 3-2.

➤ Corresponding revised parts in the manuscript:

- We revised Fig. 4a in the main manuscript to avoid the confusing expression of spin orientation. In the graph of dispersion relationship, we changed the expression of the arrows on the E_+ and E_- as Fig. R15c.

Other revised points in the main manuscript:

To clarify our intentions of each sentence, we made a minor revision in the main manuscript like below.

- At line 2 page 2
(Old) “... are too metallic, which...”
(New) “...are metallic, which...”
- At line 24 page 2
(Old) “The OMR is unprecedentedly large.”
(New) “The OMR is *found to be* unprecedentedly large.”
- At line 18 page 3
(Old) “... with same heterostructure...”
(New) “... with *the* same heterostructure...”
- At line 2 page 5
(Old) “... we found that...”
(New) “... we *find* that...”

- At line 20 page 11
 (Old) “... *samples A and B with...*”
 (New) “...*samples A and B of InAs/(Ga,Fe)Sb heterostructures with ...*”

- At the caption title of Fig. 1
 (Old) “Magnetoresistances (MRs) of InAs/(Ga,Fe)Sb bilayers”
 (New) “Magnetoresistances (MRs) of InAs/(Ga,Fe)Sb bilayer **heterostructures**”

- In the caption of Fig. 4
 (Old) “**a**, Energy dispersion...”
 (New) “**a**, Energy **band** dispersion...”

References

- R1. Moubah, R., Magnus, F., Hjørvarsson, B. & Andersson, G. Antisymmetric magnetoresistance in SmCo₅ amorphous films with imprinted in-plane magnetic anisotropy. *J. Appl. Phys.* **115**, 053911 (2014).
- R2. Xiao, C. *et al.* Linear magnetoresistance induced by intra-scattering semiclassics of Bloch electrons. *Phys. Rev. B* **101**, 201410 (2020).
- R3. Wang, Y. *et al.* Antisymmetric linear magnetoresistance and the planar Hall effect. *Nat. Commun.* **11**, 216 (2020).
- R4. Fujita, T. C. *et al.* Odd-parity magnetoresistance in pyrochlore iridate thin films with broken time-reversal symmetry. *Sci. Rep.* **5**, 9711 (2015).
- R5. Albarakati, S. *et al.* Antisymmetric magnetoresistance in van der Waals Fe₃GeTe₂/graphite/Fe₃GeTe₂ trilayer heterostructures. *Sci. Adv.* **5**, eaaw0409 (2019).
- R6. Knox, C. S., Li, L. H., Rosamond, M. C., Linfield, E. H. & Marrows, C. H. Deconvolution of Rashba and Dresselhaus spin-orbit coupling by crystal axis dependent measurements of coupled InAs/GaSb quantum wells. *Phys. Rev. B* **98**, 155323 (2018).
- R7. Goel, S., Anh, L. D., Ohya, S. & Tanaka, M. Temperature dependence of magnetic anisotropy in heavily Fe-doped ferromagnetic semiconductor (Ga,Fe)Sb. *J. Appl. Phys.* **127**, 023904 (2020).
- R8. Yuan, Y. *et al.* Nematicity of correlated systems driven by anisotropic chemical phase separation. *Phys. Rev. Mater.* **2**, 114601 (2018).
- R9. Shoenberg, D. *Magnetic Oscillations in Metals*. (Cambridge University Press, 1984). doi:10.1017/CBO9780511897870.
- R10. Saraf, S. & Rosenwaks, Y. Local measurement of semiconductor band bending and surface charge using Kelvin probe force microscopy. *Surf. Sci.* **574**, L35–L39 (2005).
- R11. Nam Hai, P., Anh, L. D. & Tanaka, M. Electron effective mass in n-type electron-induced ferromagnetic semiconductor (In,Fe)As: Evidence of conduction band transport. *Appl. Phys. Lett.* **101**, 252410 (2012).
- R12. Csonka, S. *et al.* Giant fluctuations and gate control of the g-factor in InAs nanowire quantum dots. *Nano Lett.* **8**, 3932–3935 (2008).
- R13. Cartoixà, X., Ting, D. Z.-Y. & McGill, T. C. Theoretical Investigations of Spin Splittings and Optimization of the Rashba Coefficient in Asymmetric AlSb/InAs/GaSb Heterostructures. *J. Comput. Electron.* **1**, 141–146 (2002).

Reviewers' Comments:

Reviewer #2:

Remarks to the Author:

I have read the authors' responses and I am not very satisfied with their answers for certain comments.

1. The Hall data definitely bring out the question of the non-metallic (non-Ohmic) behaviors of contact (i.e. the current dependence of OMR in the same sample has a step-like change at 200 nA) at ZERO field. Why?
2. How is the homogeneity of the InAs films before the nanofabrication? A careful AFM mapping and/or etch-pit density mapping might be needed.
3. What is the field sweeping speed during the measurement? Do the results depend on the sweeping speed?
4. Fig. 3d shows the side gating will induce such odd parity linear magnetoresistance in R14. This suggests the change of the charge density distribution will induce this effect. Therefore, in order to understand this odd behavior, a van der Pauw geometry measurements (such as those in Ref. 7) might be needed to clarify the issues.
5. The inhomogeneity of the samples may mix the Hall resistance with magnetoresistance, the authors must rule out such possibility.

The most important issue is the theoretical discussion. Generally, when field H is parallel to both σ_z and M of GaFeSb, the odd-parity MR should not be observed. So, the authors provide some new theory but I do not see their experimental results have solid link with their theory. It is very hard to prove the validity of their theory. For example, if the phenomenological parameter α depends on GaFeSb/InAs interface, the authors should show clear experimental results to support this statement.

In a word, without solid connection between theory/model and the experimental results, the paper cannot be published as its current form.

Reviewer #3:

Remarks to the Author:

The authors provided very comprehensive replies to all my questions. They have successfully addressed all my concerns. I recommend to publish the current version in Nature Communications.

Response Letter

We would like to thank the reviewers for their reports on our manuscript submitted for publication in Nature Communications (# NCOMMS-21-39056). We have carefully revised our manuscript in accordance with the comments of reviewer #2. In the following, we show point-by-point response to each comment and how we revised our main manuscript and Supplementary Information.

Reviewer #2:

Comment 1

1. The Hall data definitely bring out the question of the non-metallic (non-Ohmic) behaviors of contact (i.e. the current dependence of OMR in the same sample has a step-like change at 200 nA) at ZERO field. Why?

Our response:

We thank the reviewer for the insightful comment.

Firstly, we would like to point out that the jumps in our Hall resistance data near zero magnetic field (see Fig. R1a) are artifacts from our lock-in measurements, not intrinsic properties of the sample. Close to zero magnetic field, when the Hall voltage switches from positive (negative) to negative (positive), the small output voltage makes the phase-offset unstable. This causes the jumps in the Hall resistance data near zero magnetic field in the AC lock-in measurements. For comparison, one can see that the Hall resistance data measured using a DC current does not show such jumps, as shown in Fig. R1b.

Secondly, we note that the Hall resistance and the odd-parity magnetoresistance (OMR) values in our discussion were obtained at high magnetic field ($B \sim 1$ T), where the abovementioned artifacts are irrelevant. Within these reliable values at high magnetic field, the Hall resistances are unchanged regardless of the current magnitude as expected. However, the longitudinal resistance R_0 and the OMR values at 1 T clearly show a jump at 200 nA in the current (I) dependence, as shown in Fig. R2. This different current dependence indicates that the Hall resistance and the OMR have different origins.

Thirdly, it is well known that in InAs the Fermi energy pinning position is above the conduction band bottom at its interface [see for example, C. A. Mead and W. G. Spitzer, Phys. Rev. Lett. **10**, 472 (1963) and L. F. J. Piper et al., Phys. Rev. B **73**, 195321 (2006)]. Therefore, the interface between a metal electrode and an n-type InAs forms an Ohmic contact. Thus, non-Ohmic behavior caused by the Au/InAs contacts in our device is very unlikely; we actually confirmed Ohmic contacts in our devices.

As already explained in the previous manuscript, the reason why the OMR shows the step-like change against the current I , as shown in Fig. R2, can be attributed to the coexistence of one- and two-dimensional transport paths in the InAs channel. We briefly summarize it in the following.

In the InAs channel, there are parallel conduction in the edge (one-dimensional (1D)) and center (two-dimensional (2D)) channels (see Fig. R3a). In both 1D and 2D

channels, there are magnetic proximity effects (MPE) induced by the perpendicular magnetization component M_z of the underlying (Ga,Fe)Sb, as presented in our previous work¹. However, we expect that the MPE occurs more strongly in the center 2D channel than in the edge 1D channels. This is because the M_z component is smaller in the edges of (Ga,Fe)Sb where the magnetic moments of Fe usually tilt towards the side surface. Therefore, we think that a possible reason for the step-like increase of OMR at $I_C = 200$ nA is sudden enhancement of the MPE in the edge due to the expansion of the electron wavefunctions in the 1D edge channel towards the 2D center channel at this critical current value.

As illustrated in Fig. R3b, in the 1D edge channel, the electron wavefunction is confined by the triangular potential at the side surfaces and its penetration to the 2D center channel is very small at $I < I_C$. When we increase I , however, the current is more concentrated in the edge, which has higher conductivity because of weaker magnetic scattering from MPE. This increases the electron carrier concentration in the edge. At $I_C < I$, these changes may eventually lead to the occupation of the next quantized level at a slightly higher energy, whose electron wavefunction overlaps more largely with the 2D channel due to the weaker confinement. This enhances the 1D (edge) - 2D (center) wavefunction overlapping and consequently increases the MPE in the edge channels in a sudden manner as observed at $I_C = 200$ nA, leading to the sudden increase in $\Delta R/R_0$ (from 0.1 to 0.3 %), as shown in Fig.R3b. With more magnetic scattering in the edge transport, this also explains the slight increase of the total resistance R_0 (from ~ 3.00 to ~ 3.05 k Ω) at I_C in Fig. R3b.

➤ **Corresponding revised parts in the manuscript:**

In order to explain the behaviour of our Hall data near zero field,

- We added the following sentences to Supplementary Note 5: “*Here, we note that... we actually confirmed Ohmic contacts in our devices.*”
- We added Fig. R1b as Supplementary Fig. 9b.
- We added the citations C. A. Mead and W. G. Spitzer, Phys. Rev. Lett. **10**, 472 (1963) and L. F. J. Piper et al., Phys. Rev. B **73**, 195321 (2006) to Supplementary Note 5.

Fig. R1 Hall resistance data in InAs/(Ga,Fe)Sb measured with (a = Supplementary Fig. 9) AC and (b) DC current at 3.5 K. When we measured the Hall resistance with DC current, the jumps near zero field does not appear. We note that the difference of the magnitude of Hall resistance between a and b is due to the difference of the samples.

Fig.R2 a, Current dependence of the magnetoresistance (upper panel) and the odd component (lower panel) at 3.5 K with perpendicular B . Note that $R_{23}^{\text{odd}}(B) = (R_{23}(B) - R_{23}(-B))/2$. Here, $R_{23}(B)$ is the resistance measured between terminals 2 and 3, and R_0 is $R_{23}(0 \text{ T})$. **b**, Current dependence of R_0 (upper panel) and the OMR magnitude $\Delta R/R_0 = R_{23}^{\text{odd}}(1 \text{ T})/R_0$ (lower panel).

Fig. R3 a, Schematic illustration of the InAs/(Ga,Fe)Sb bilayer heterostructure near the Hall bar edge. In the (Ga,Fe)Sb layer, magnetic moments M_z' (red arrows) near the side edge may be canted and do not effectively induce the MPE in the InAs edge channel. On the other hand, in the 2D channel (center) side, magnetic moments M_z are aligned in the z direction due to the perpendicular magnetic anisotropy of (Ga,Fe)Sb, leading to strong MPE. Through overlapping of electron wavefunctions in the 1D channel (edge) with the 2D channel (center), MPE is strongly induced in the 1D channel at $I > I_c$. The MPE, together with the Rashba SOI, leads to the appearance of OMR. **b**, Illustrated electronic subband structure of the conduction band bottom $E_{C(\text{InAs})}$ of InAs near the edge (blue curves). When the current I is increased, the electron carriers are accumulated near the edge. This leads to a change in the occupied quantized levels at I_c . When higher levels are occupied by electrons at $I > I_c$, the 1D wavefunctions largely overlap with the 2D center region, which suddenly enhances the MPE in the edge, leading to the sudden increase of OMR.

Comment 2

How is the homogeneity of the InAs films before the nanofabrication? A careful AFM mapping and/or etch-pit density mapping might be needed.

Our response:

Figure R4 shows the atomic force microscope (AFM) images of the InAs/(Ga,Fe)Sb sample before (left) and after (right) etching using Ar ion milling, similar to the fabrication process of the field effect transistors in our work. The root mean square (RMS) of the roughness is only 0.4 nm, which is less than two monolayers of InAs, and remains almost unchanged after the etching. Also, no apparent etch pit is observed in these AFM images. This result indicates that our films are homogeneous and smooth, which is not affected by the device fabrication process.

Fig. R4 Surface morphology measured by atomic force microscopy (AFM) of the InAs/(Ga,Fe)Sb heterostructure before (left) and after (right) etching using Ar ion milling.

➤ Corresponding revised parts in the manuscript:

In order to clarify that the inhomogeneity effect on our OMR is negligible,

- We added Fig. R4 as Supplementary Fig. 10a.
- We added the following sentences in Method section "Fabrication process of the Hall bar devices and transport measurement": "*Supplementary Figure 10a shows... by the device process.*"

Comment 3

What is the field sweeping speed during the measurement? Do the results depend on the sweeping speed?

Our response:

We conducted the measurement of OMR using different systems, a PPMS with a superconducting magnet ($|B| < 14$ T) and a self-designed system with an electromagnet ($|B| < 1$ T). The magnetic field sweeping speed in the former (PPMS) was typically 0.3 T/min,

while that in the latter was roughly 0.1 T/min. The OMR data obtained in these two machines are in good agreement as shown in Fig. R5a. In addition, the OMR exists in our gate-voltage sweeping experiment with a *constant* magnetic field as shown in Fig. R5b. Thus, the OMR data are not affected by changing the measurement system or the magnetic field sweeping speed.

➤ **Corresponding revised parts in the manuscript:**

In order to clarify that magnetic sweeping does not affect our OMR,

- We added Fig. R5a as Supplementary Fig. 16.
- We added new Supplementary Note 10 “Magnetic field sweeping effect on OMR”.

Fig. R5 a, Comparison of odd components of magnetoresistance ($R_{23}^{\text{odd}} = [R_{23}(B) - R_{23}(-B)]/2$, B is the magnetic field.) measured by two different transport measurement systems with electromagnet (red) and superconducting magnet (blue). These data are measured at 3.5 K and 2 K, respectively with 1 μA . B is applied perpendicular to the plane. **b**, (= Fig. 3c in the main manuscript) Gate voltage V_g dependence of R_{23} at various B of -10 T (green), 0 T (white), 10 T (orange) (upper panel), and that of the odd component $\Delta R/R_0$ (lower panel), where $\Delta R = (R_{23}(-10 \text{ T}) - R_{23}(+10 \text{ T}))/2$, and $R_0 = R_{23}(0 \text{ T})$, measured at 2 K on device D1. These measurements are conducted with a fixed current of 1 μA at 2 K.

Comment 4

Fig. 3d shows the side gating will induce such odd parity linear magnetoresistance in R14. This suggests the change of the charge density distribution will induce this effect. Therefore, in order to understand this odd behavior, a van der Pauw geometry measurements (such as those in Ref. 7) might be needed to clarify the issues.

Our response:

Figure R6 shows our magnetoresistance data obtained by the van der Pauw method^{2,3}. We made four contacts (A, B, C, and D) on the top surface of a 1.3 mm \times 3.0 mm InAs/(Ga,Fe)Sb sample with indium paste, as shown in Fig. R6a. Let $R_{AB,CD}$ ($R_{BC,DA}$) be $V_{CD(DA)}/I_{AB(BC)}$, where $V_{CD(DA)}$ is the voltage between C(D) and D(A) with the current $I_{AB(BC)}$ flowing between A(B) and B(C). In the van der Pauw method, the resistivity ρ is given by

Fig. R6 a, Experimental setup of our van der Pauw method. Blue and grey areas indicate the surface of our sample (InAs/(Ga,Fe)Sb) and indium paste. **b**, Variation of resistivity ρ (upper) and its odd components ρ_{odd} (lower) against the magnetic field B measured by van der Pauw method at 3.5 K with 1 μ A. B is perpendicular to the plane.

$$\rho = \frac{\pi d}{\ln 2} \frac{R_{AB,CD} + R_{BC,DA}}{2} f\left(\frac{R_{AB,CD}}{R_{BC,DA}}\right) \quad (\text{R1})$$

where d is the thickness of the sample, and f is a function satisfying

$$\frac{R_{AB,CD} - R_{BC,DA}}{R_{AB,CD} + R_{BC,DA}} = \frac{f}{\ln 2} \operatorname{arccosh}\left[\exp\left(\frac{\ln 2}{f}\right)/2\right]. \quad (\text{R2})$$

As shown in Fig. R6b, the OMR magnitude at 1 T and 1 μA is 0.2%, which is the same order as that measured in the Hall bar geometry. The OMR in our InAs/(Ga,Fe)Sb appears regardless of the measurement configuration or device process. However, since the van der Pauw method is only valid and available in the system with uniform current distribution, it is difficult to analyze and explain the data obtained in this measurement. This is because the 1D and 2D current paths coexist in our InAs/(Ga,Fe)Sb.

Comment 5

The inhomogeneity of the samples may mix the Hall resistance with magnetoresistance, the authors must rule out such possibility.

Our response:

We show the Hall resistance data of device D1 ($= V_{26}/I_{14}$) in Fig. R7 (same data are shown in Supplementary Fig. 9 in our Supplementary Information). From Fig. R7a, one can see that the Hall resistances are independent of the current I in the whole range from 100 nA to 100 μA . In contrast, the current dependence of OMR in the same sample has a step-like change at 200 nA, as shown in Fig. R7b. This fact indicates that the origin of the OMR is completely different from that of the Hall effect. In addition, we also investigate the Hall resistance in device D2 with separate gate electrodes (Fig. R8a) with applying a separated gate voltage V_{g1} . Figure R8b shows the Hall resistance at various V_{g1} , where V_{g1} can change the contribution of the 1D edge channel transport in InAs. Again, as shown in Fig. R8b and c, the evolution of the OMR and the Hall resistance with applying the gate voltage are

Fig R7 a, (= Supplementary Fig. 9a) Hall resistance (V_{26}/I_{14}) of device D1 at various current measured by a lock-in technique at a frequency of 5261 Hz at 3.5 K. **b**, (= Fig. 2a in the main manuscript) Current (I) dependence of the OMR magnitude $\Delta R/R_0$ ($= [(R_{23}(1 \text{ T}) - R_{23}(-1 \text{ T}))/2]/R_0$) measured in R_{23} at 3.5 K. R_0 is the zero field resistance. The R_0 and $\Delta R/R_0$ jump up at I_c ($= 200 \text{ nA}$) simultaneously (see Fig. 2a in the main manuscript).

completely different: The OMR in a two-terminal measurement (odd component R_{14}^{odd} of $R_{14} = V_{14}/I_{14}$) exhibits the polarity change with V_{g1} (lower panel of Fig. R8c), while the Hall resistance (V_{26}/I_{14}) shows a negative slope in all V_{g1} (Fig. R8b). From these results, we can rule out the possibility of Hall resistance - magnetoresistance mixing as an origin of the OMR.

Fig R8 a, (= Fig. 3b in the main manuscript) Optical microscopy image of the Hall-bar field-effect transistor (FET) device D2 with separated gate electrodes G_1 and G_2 , respectively. The dashed line indicates the outline of the Hall bars. The light and dark yellow parts are the Au pads of gate and Hall-bar electrodes, respectively. **b**, (= Supplementary Fig. 9b) Hall resistance (V_{26}/I_{14}) vs. perpendicular magnetic field B of device D2 measured at various gate voltage V_{g1} (applied to gate G_1) at 2 K with $1 \mu\text{A}$. **c**, (= Fig. 3d in the main manuscript) MR results of device D2 (top panel) and the odd components (bottom panel) at 2 K. R_{14} is the resistance measured between terminals 1 and 4 (two-terminal measurement). The MR results at $V_{g1} = -7, 0, +7 \text{ V}$ are shown in green, black and blue lines, respectively.

Comment 6

The most important issue is the theoretical discussion. Generally, when filed H is parallel to both σ_z and M of GaFeSb, the odd-parity MR should not be observed. So, the authors provide some new theory but I do not see their experimental results have solid link with their theory. It is very hard to prove the validity of their theory. For example, if the phenomenological parameter α depends on GaFeSb/InAs interface, the authors should show clear experimental results to support this statement.

Our response:

There are two unique aspects of our OMR that make it stand out of all the previously reported \mathbf{B} -odd magnetoresistances (see Table R1):

1. Our OMR does not depend on the current direction. This indicates that it is not a non-reciprocal transport phenomenon, which was observed in the non-linear response region⁴⁻⁷.
2. Our OMR appears even when the magnetic field \mathbf{B} is parallel to the magnetization \mathbf{M} ($\mathbf{B} // \mathbf{M}$). This is completely different from other linear MR phenomena⁸⁻¹² (observed in the linear response region), where \mathbf{B} and \mathbf{M} must be non-collinear.

Thus, the theoretical models proposed thus far to describe the \mathbf{B} -odd components of magnetoresistance such as the unidirectional MR (observed in the non-linear response region)⁴⁻⁷ and linear MR⁸⁻¹² (observed in the linear response region) cannot explain the new OMR observed in our study. Therefore, a new theoretical model is definitely required.

Furthermore, there are two important points deduced directly from our experimental results and careful considerations on the mechanism of our OMR:

- First, the coupling between the edge-center channels plays a vital role in inducing the new OMR, which is deduced from the following experimental facts:
 - The OMR occurs at the edge transport of the InAs layer, which is evident from its absence in the two-terminal measurement and its opposite signs when measured along the opposite edges, as shown in Fig. 2 and discussed in the main manuscript.
 - The OMR requires a magnetic proximity effect from (Ga,Fe)Sb to InAs, as demonstrated by the global gate control experiment shown in Fig. 3a,c of the main manuscript. The magnetic coupling at the (Ga,Fe)Sb/InAs interface, however, occurs mainly in the center region (2D channel) of the InAs layer. This is evident in the absence and appearance of a Kondo-like tail at $T < \sim 10$ K in the temperature dependence of the resistance of the edge (1D) and center (2D) channels, respectively, as shown in Fig. R9 (see also Supplementary Note 6 in Supplementary Information).
- Second, OMR will appear if there is asymmetry in the scattering rate of electron carriers in the two bands E_+ and E_- of the energy-momentum dispersion relationship (see Fig. R10 = Fig. 4b in the main manuscript). This was straightforwardly deduced from the 1D Hamiltonian of the edge channel and the Boltzmann's formalism discussed in the Method section. As shown in Fig. R10, it is also important to point out that these two bands (E_+ and E_-) are characterized by a spin-momentum locking effect with opposite chiralities, due to the Rashba spin-orbit coupling in the InAs edge channel.

We emphasize that all these features are straightforwardly deduced from the experimental results and the Hamiltonian of the InAs edge channel, without any unreasonable assumption.

Therefore, in our proposed model, we figure out that there will be asymmetry in the scattering rate of electron carriers in the two bands (E_+ and E_-) if one considers spin-momentum locking effect in both the edge and center channels, which depends only on the current direction. Here, we made only one assumption that scattering events between the two channels must occur within the same chirality, which is totally reasonable. This chirality

reservation in the edge-center scattering of InAs yields asymmetric scattering rates ($\tau_+ \neq \tau_-$, where τ_{\pm} indicates the relaxation time for the scattering between the 1D – 2D channels of the E_+ and E_- bands), as illustrated in Fig. R10. From this assumption, our theoretical model expects that

(i) OMR in InAs/(Ga,Fe)Sb emerges even when $\mathbf{B} // \mathbf{M}$.

(ii) OMR in InAs/(Ga,Fe)Sb is not affected by reversing the current direction.

(iii) OMR in InAs/(Ga,Fe)Sb can be modulated by varying the magnetic proximity effect. These notable features are confirmed by our experimental results [(i) Fig. 1c in the main manuscript, (ii) Supplementary Fig. 4, (iii) Fig. 3c in the main manuscript. See Supplementary Note 8 for the detailed discussion].

At this stage, our model based on the chirality-dependent scattering between the edge (1D) and center (2D) channels is the first theoretical model that can explain all the aspects of this new OMR. By introducing the chirality-dependent scattering into our Boltzmann formalism, our theoretical model successfully reproduces the linear change against \mathbf{B} and magnitude of OMR (13.5% at 10 T, see Fig. R11 = Figs. 4c,d,e in the main manuscript). This shows one of the direct links between our experimental results and our theoretical model.

We acknowledge that it is challenging to directly verify the chirality-dependent scattering from the present experimental setup, as commented by the reviewer, because the spin-orbit coupling cannot be changed independently of other parameters such as carrier density or magnetic proximity effect. However, given the uniqueness and novelty, as well as the possibility of device applications of our observed new OMR, we think that our paper is compelling and it is expected that more sophisticated theories will follow in the near future. We hope that the reviewer shares this view with us.

➤ **Corresponding revised parts in the manuscript:**

In order to discuss the issue described above, we revised the manuscript as follows:

- We added new Supplementary Note 9 “Correspondence between our theoretical model and experimental results”.
- To mention the discussion in Comment 6, we revised line 23 page 9 in the main manuscript like below:
 - (old) *(See Supplementary Note 8 for...*
 - (new) *(See Supplementary Note 8 and 9 for...*
- We added the citations #4 - #10, and #12 in this letter to Supplementary Note 9.

Table R1 (= Supplementary Table 1). Comparison of OMR observed in previous reports and our work. The maximum OMR magnitude $\Delta R/R_0$ ($= [(R(B)-R(-B))/2]/R(0 \text{ T})$) normalized by R_0 ($= R(0 \text{ T})$) were obtained under magnetic field B at temperature T .

Material	$\Delta R/R_0$ (%)	B (T)	T (K)	Observable under $B // M$	Proposed origin	Refs.
SmCo ₅	1.3×10^{-2}	0.015	room temp.	No	non-uniform distribution of the magnetization	R13
SmCo ₅	4.6×10^{-2}	0.5	300	No	Zeeman splitting/ anomalous Hall effect	R11, R14
Gd ₂ Os ₂ O ₇	5.0×10^{-2}	2	195	No	magnetic domain walls	R14
Eu ₂ Ir ₂ O ₇ (theory)	-	-	-	No	Berry curvature, magnetic moment, and shift vector	R11
Eu ₂ Ir ₂ O ₇ (experiment)	0.44	9	2	No	magnetic texture	R15
Fe ₃ GeTe ₂ / graphite/ Fe ₃ GeTe ₂	1.1	0.01	50	No	interfacial SOI of Fe ₃ GeTe ₂ as a topological nodal line	R16
InAs/ (Ga,Fe)Sb	13.5 5	10 10	2 300	Yes	Rashba SOI at the edge of InAs and magnetic proximity effect	Our work

Fig. R9 (= Supplementary Fig. 11a-c) **a**, Optical microscope image (same as Fig. 1b in the main manuscript) of the Hall bar device D_L . Here, w and l_{23} indicate the width and the distance between “2” and “3” electrodes, respectively. **b**, Temperature (T) dependence of $R_{23}(0 T)$ of device D_L (blue) and device D_S (green). **c**, T dependence of R_{2D} (cyan) and R_{1D} (purple) in D_L . The inset shows the schematic resistor network representing R_{23} . The 2D resistance R_{2D} has the width w and length l_{23} , and the 1D resistance R_{1D} has the same length. The black dashed line at $T < \sim 10 \text{ K}$ is the fitting result using a logarithmic function, which is characteristic of the Kondo effect ($R_{2D} = R_{c0} - R_{c1} \ln T$, where R_{c0} and R_{c1} are fitting parameters).

Fig. R10 (= Fig. 4b in the main manuscript), Schematic energy dispersion (upper) and its Fermi surface (lower) of the 1D (right) and 2D (left) channels. Purple dashed line indicates the Fermi level E_F . Due to the Rashba SOI in the z direction in both channels, the y spin component (σ_y) of electrons is locked to the momentum k_x in opposite directions between the green and pink bands. Electron scattering between the 1D and 2D channels occurs mainly between bands with the same chirality, as indicated by the red and blue arrows. In the 2D channel, MPE opens the gap (Δ_{2D}) between two bands with opposite z spin components (σ_z). Different density of states between the two (pink and green) 2D bands leads to different relaxation times τ_+ and τ_- in the 1D channel.

Fig. R11 (= Fig. 4c-e in the main manuscript) **a**, Calculated results of the OMR using eq. (4) in the main manuscript with asymmetric scattering rate $\alpha = 0.1$, $E_F = 100$ meV, $m^* \lambda_{\text{side}}^2 = 0.45$ meV¹⁷, and $\Delta_z/B_z = 0.52$ meV/T for $g = 18$ ¹⁸. The sign of the Rashba parameter λ_{side} determines the polarity of the OMR component in the 1D system. **b**, and **c**, OMR as functions of α (with $E_F = 100$ meV) and E_F (with $\alpha = 0.1$), respectively.

Other revised points in the main manuscript:

- We added a co-author Harunori Shiratani, who helped the sample preparation and AFM measurement for this response letter.
- We corrected the wrong numbering of eq. (4) in Supplementary Note 7 and 8. (wrong: eq.(5), correct: eq.(4))
- We corrected the value of y-axis of Fig. 3c in the main manuscript, which was mistakenly written without a decimal separator (a period “.”), leading to all values multiplied by 10.
- We corrected the definition of the asymmetric scattering rate α (line 2 page 10 in the main manuscript and eq. (S20) in Supplementary Note 7):

$$\text{(old) } \alpha = \tau_+/\tau_-$$

$$\text{(new) } \alpha = \tau_-/\tau_+$$

Although we had to change this point in the previous revision of Supplementary Note 8 to maintain the consistency of the whole manuscript, we overlooked it. We would like to apologize for this oversight. Due to this change, the polarity of OMR expressed in eq. (4) is inversed. However, all the discussion regarding α remains consistent, and thus, our conclusions are not changed at all.

Following this modification, we make some revisions for our theoretical part:

- ◆ eq. (4) in the main manuscript
(old)

$$\sigma_{xx} \simeq \frac{e^2}{h} \tau_- |\lambda_{\text{side}}| \sqrt{1 + \frac{2E_F}{m^* \lambda_{\text{side}}^2}} \left[1 + \alpha + (1 - \alpha) \frac{|\lambda_{\text{side}}|}{\lambda_{\text{side}}} \frac{g\mu_B}{2E_F + m^* \lambda_{\text{side}}^2} B_z \right]$$

(new)

$$\sigma_{xx} \simeq \frac{e^2}{h} \tau_+ |\lambda_{\text{side}}| \sqrt{1 + \frac{2E_F}{m^* \lambda_{\text{side}}^2}} \left[1 + \alpha - (1 - \alpha) \frac{|\lambda_{\text{side}}|}{\lambda_{\text{side}}} \frac{g\mu_B}{2E_F + m^* \lambda_{\text{side}}^2} B_z \right]$$

◆ Fig. 4b-e in the main manuscript (old)

(new)

- ◆ eq. (S18) in Supplementary Note 7
(old)

$$C_1 \tau_- |\lambda_{\text{side}}| \left[\left(1 + \frac{2E_F}{m^* \lambda_{\text{side}}^2} + \frac{|\lambda_{\text{side}}|}{\lambda_{\text{side}}} \frac{2\Delta_z}{m^* \lambda_{\text{side}}^2} \right)^{\frac{1}{2}} + \alpha \left(1 + \frac{2E_F}{m^* \lambda_{\text{side}}^2} - \frac{|\lambda_{\text{side}}|}{\lambda_{\text{side}}} \frac{2\Delta_z}{m^* \lambda_{\text{side}}^2} \right)^{\frac{1}{2}} \right]$$

(new)

$$C_1 \tau_+ |\lambda_{\text{side}}| \left[\left(1 + \frac{2E_F}{m^* \lambda_{\text{side}}^2} - \frac{|\lambda_{\text{side}}|}{\lambda_{\text{side}}} \frac{2\Delta_z}{m^* \lambda_{\text{side}}^2} \right)^{\frac{1}{2}} + \alpha \left(1 + \frac{2E_F}{m^* \lambda_{\text{side}}^2} + \frac{|\lambda_{\text{side}}|}{\lambda_{\text{side}}} \frac{2\Delta_z}{m^* \lambda_{\text{side}}^2} \right)^{\frac{1}{2}} \right]$$

- ◆ eq. (S19) in Supplementary Note 7
(old)

$$J_x \simeq C_1 \tau_- |\lambda_{\text{side}}| \sqrt{1 + \frac{2E_F}{m^* \lambda_{\text{side}}^2}} \left[1 + \alpha + (1 - \alpha) \frac{|\lambda_{\text{side}}|}{\lambda_{\text{side}}} \frac{\Delta_z}{2E_F + m^* \lambda_{\text{side}}^2} \right]$$

(new)

$$J_x \simeq C_1 \tau_+ |\lambda_{\text{side}}| \sqrt{1 + \frac{2E_F}{m^* \lambda_{\text{side}}^2}} \left[1 + \alpha - (1 - \alpha) \frac{|\lambda_{\text{side}}|}{\lambda_{\text{side}}} \frac{\Delta_z}{2E_F + m^* \lambda_{\text{side}}^2} \right]$$

- ◆ the caption of Fig. 4 in the main manuscript
(old) α represents the strength of the MPE at the interface; ...
(new) α (= τ_-/τ_+) represents the different relaxation time of electron carriers in the E^- and E^+ states; ...

◆ Supplementary Fig. 15
(old)

(new)

References

1. Takiguchi, K. *et al.* Giant gate-controlled proximity magnetoresistance in semiconductor-based ferromagnetic–non-magnetic bilayers. *Nat. Phys.* **15**, 1134–1139 (2019).
2. van der Pauw, L. J. A Method of Measuring the Resistivity and Hall Coefficient on Lamellae of Arbitrary Shape. *Philips Tech. Rev.* **20**, 220–224 (1958).
3. van der Pauw, L. J. A method of measuring specific resistivity and Hall effect of discs of arbitrary shape. *Philips Res. Reports* **13**, 1–9 (1958).
4. Tokura, Y. & Nagaosa, N. Nonreciprocal responses from non-centrosymmetric quantum materials. *Nat. Commun.* **9**, 3740 (2018).
5. Zhang, S. S. L. & Vignale, G. Theory of unidirectional spin Hall magnetoresistance in heavy-metal/ferromagnetic-metal bilayers. *Phys. Rev. B* **94**, 140411(R) (2016).
6. Rikken, G. L. J. A., Fölling, J. & Wyder, P. Electrical magnetochiral anisotropy. *Phys. Rev. Lett.* **87**, 236602 (2001).
7. Rikken, G. L. J. A. & Wyder, P. Magnetoelectric anisotropy in diffusive transport. *Phys. Rev. Lett.* **94**, 016601 (2005).
8. Singha, R., Satpati, B. & Mandal, P. Fermi surface topology and signature of surface Dirac nodes in LaBi. *Sci. Rep.* **7**, 6321 (2017).
9. Abrikosov, A. A. Quantum linear magnetoresistance; solution of an old mystery. *J. Phys. A: Math. Gen.* **36**, 9119–9131 (2003).
10. Abrikosov, A. Quantum magnetoresistance. *Phys. Rev. B* **58**, 2788–2794 (1998).
11. Xiao, C. *et al.* Linear magnetoresistance induced by intra-scattering semiclassics of Bloch electrons. *Phys. Rev. B* **101**, 201410 (2020).
12. Zyuzin, V. A. Linear magnetoconductivity in magnetic metals. *Phys. Rev. B* **104**, L140407 (2021).
13. Moubah, R., Magnus, F., Hjörvarsson, B. & Andersson, G. Antisymmetric magnetoresistance in SmCo₅ amorphous films with imprinted in-plane magnetic anisotropy. *J. Appl. Phys.* **115**, 053911 (2014).
14. Wang, Y. *et al.* Antisymmetric linear magnetoresistance and the planar Hall effect. *Nat. Commun.* **11**, 216 (2020).
15. Fujita, T. C. *et al.* Odd-parity magnetoresistance in pyrochlore iridate thin films with broken time-reversal symmetry. *Sci. Rep.* **5**, 9711 (2015).
16. Albarakati, S. *et al.* Antisymmetric magnetoresistance in van der Waals Fe₃GeTe₂/graphite/Fe₃GeTe₂ trilayer heterostructures. *Sci. Adv.* **5**, eaaw0409 (2019).
17. Cartoixà, X., Ting, D. Z.-Y. & McGill, T. C. Theoretical Investigations of Spin Splittings and Optimization of the Rashba Coefficient in Asymmetric AlSb/InAs/GaSb Heterostructures. *J. Comput. Electron.* **1**, 141–146 (2002).
18. Csonka, S. *et al.* Giant fluctuations and gate control of the g-factor in InAs nanowire quantum dots. *Nano Lett.* **8**, 3932–3935 (2008).

Reviewers' Comments:

Reviewer #2:

Remarks to the Author:

The authors have carefully responded my comments in detail. Most of the responses are very impressive and convincing. Although I am still not fully convinced in some issues, specifically in theory part, I believed that the experimental setup and results are well presented now and the phenomena found in this work is indeed interesting and is necessary to be further investigated. So I am glad to recommend the publication of the paper in its current form and I hope this publication may bring interest to the researchers in the various fields. In this case, a debate of the origin of the odd linear resistances is encouraging.

Response Letter

We would like to thank the reviewer for recommending acceptance of our manuscript submitted for publication in Nature Communications (# NCOMMS-21-39056). This is our last response for the comment from Reviewer #2.

Reviewer #2:

Comment

The authors have carefully responded my comments in detail. Most of the responses are very impressive and convincing. Although I am still not fully convinced in some issues, specifically in theory part, I believed that the experimental setup and results are well presented now and the phenomena found in this work is indeed interesting and is necessary to be further investigated. So I am glad to recommend the publication of the paper in its current form and I hope this publication may bring interest to the researchers in the various fields. In this case, a debate of the origin of the odd linear resistances is encouraging.

Our response:

We appreciate the reviewer's high evaluation of our results and recommendation for publication in *Nature Communications*. As the reviewer pointed out, there is still room for improvement regarding the theoretical part about chirality dependent scattering. We hope that this paper will stimulate further research including this theoretical part, and we would like to make further efforts in this field, particularly deeper understanding and better controlling the novel magnetoresistance.

Other revised points in the main manuscript:

1. We revised Fig. 1a, changed all the vectors (Current **I**, magnetic field **B**, electric field **E_{sur}**) into bold style. In the graph, the "**E_{sur}**" was corrected to "**-eE_{sur}**", which is the force that is exerted on the electron carriers in the edge channel.
2. Correspondingly, we revised the following expression in page 6, line 12-13 of the main manuscript:
Old: "the built-in electric field **E_{sur}** points outward along the y direction"
New: "the built-in electric field **E_{sur}** pushes the electron carriers outward along the y direction"

These corrections are to improve the readability of the manuscript and do not affect the conclusions of our work.